# Structure of the OMEGA nickase IsrB in complex with ωRNA and target DNA

Seiichi Hirano[1,2,3,4,5,8], Kalli Kappel[1,2,3,4,5,8], Han Altae-Tran[1,2,3,4,5], Guilhem Faure[1,2,3,4,5], Max E. Wilkinson[1,2,3,4,5], Soumya Kannan[1,2,3,4,5], F. Esra Demircioglu[1,2,3,4,5], Rui Yan[6], Momoko Shiozaki[6], Zhiheng Yu[6], Kira S. Makarova[7], Eugene V. Koonin[7], Rhiannon K. Macrae[1,2,3,4,5] & Feng Zhang[1,2,3,4,5 ✉]

RNA-guided systems, such as CRISPR–Cas, combine programmable substrate recognition with enzymatic function, a combination that has been used advantageously to develop powerful molecular technologies[1,2]. Structural studies of these systems have illuminated how the RNA and protein jointly recognize and cleave their substrates, guiding rational engineering for further technology development[3]. Recent work identified a new class of RNA-guided systems, termed OMEGA, which include IscB, the likely ancestor of Cas9, and the nickase IsrB, a homologue of IscB lacking the HNH nuclease domain[4]. IsrB consists of only around 350 amino acids, but its small size is counterbalanced by a relatively large RNA guide (roughly 300-nt ωRNA). Here, we report the cryogenic-electron microscopy structure of *Desulfovirgula thermocuniculi* IsrB (DtIsrB) in complex with its cognate ωRNA and a target DNA. We find the overall structure of the IsrB protein shares a common scaffold with Cas9. In contrast to Cas9, however, which uses a recognition (REC) lobe to facilitate target selection, IsrB relies on its ωRNA, part of which forms an intricate ternary structure positioned analogously to REC. Structural analyses of IsrB and its ωRNA as well as comparisons to other RNA-guided systems highlight the functional interplay between protein and RNA, advancing our understanding of the biology and evolution of these diverse systems.

The RNA-guided IsrB protein is an OMEGA family member encoded in the IS200/IS605 superfamily of transposons. IsrB is the likely antecedent of IscB, another OMEGA family member that is the apparent ancestor of Cas9, as indicated both by phylogenetic analysis and by the shared unique domain architecture[4,5]. Like IscB and Cas9, IsrB contains a RuvC-like nuclease domain that is interrupted by the insertion of a bridge helix (BH) (Fig. 1a). However, in contrast to IscB and Cas9, IsrB lacks the HNH nuclease domain, the REC lobe and large portions of the protospacer adjacent motif- (PAM-)interacting domain and, accordingly, is much smaller (at roughly 350 amino acids) than Cas9. IsrB additionally contains an N-terminal PLMP domain (named after its conserved amino acid motif) and an uncharacterized C-terminal domain (Fig. 1b). Previous work has shown that IsrB associates with a roughly 300-nt ωRNA, which guides IsrB to nick the non-target strand of double-stranded (ds) DNA containing a 5′-NTGA-3′ target-adjacent motif (TAM)[4].

## Structure of the IsrB–ωRNA-target DNA complex

To characterize the molecular mechanism of ωRNA-guided DNA targeting by IsrB, we analysed a ternary complex comprising *Desulfovirgula thermocuniculi* IsrB (DtIsrB), a 284-nt ωRNA containing a 20-nt guide

segment, a 31-nt target DNA strand and a 10-nt non-target DNA strand using single-particle cryo-EM (Fig. 1c). We obtained a three-dimensional (3D) reconstruction of the ternary complex with an overall resolution of 3.1 Å (Fig. 1d, Extended Data Fig. 1a–c and Extended Data Table 1). Some regions of the map corresponding to the ωRNA, however, were resolved at a lower resolution. To refine the modelling of the RNA coordinates, we used an RNA-specific modelling tool, auto-DRRAFTER, together with a covariance-based secondary structure model to build an initial ωRNA model. On the basis of this ωRNA model and an initial IsrB model generated by protein structure prediction, we determined the IsrB–ωRNA–DNA structure (Fig. 1e and Extended Data Figs. 1d,e and 2)[6–8].

The structure revealed that IsrB extensively binds to target DNA through a 20-nt duplex between the ωRNA and target DNA (Fig. 1e). The RuvC domain (residues 60–253) encompasses the three catalytic motifs (RuvC I–III) and three insertions (BH (residues 92–112), A (residues 113–129) and B (residues 161–179)) (Fig. 1b). Insertion A is a 'shortcut' linker between BH and RuvC II; this linker is replaced with the REC lobe in Cas9. Thus, we denote this insertion the REC linker (RECL). Insertion B, between RuvC II and III, is a simple linker consisting of a loop and an α helix that in the IsrB structure occupies a position corresponding to that of the HNH domain in Cas9. Thus, we denote it the HNH linker (HNHL). The C-terminal domain (residues 287–351) adopts a core fold

[1]Broad Institute of MIT and Harvard, Cambridge, MA, USA. [2]McGovern Institute for Brain Research at MIT, Cambridge, MA, USA. [3]Department of Brain and Cognitive Science, Massachusetts Institute of Technology, Cambridge, MA, USA. [4]Department of Biological Engineering, Massachusetts Institute of Technology, Cambridge, MA, USA. [5]Howard Hughes Medical Institute, Cambridge, MA, USA. [6]CryoEM Shared Resources, Howard Hughes Medical Institute, Janelia Research Campus, Ashburn, VA, USA. [7]National Center for Biotechnology Information, National Library of Medicine, National Institutes of Health, Bethesda, MD, USA. [8]These authors contributed equally: Seiichi Hirano, Kalli Kappel. ✉e-mail: zhang@broadinstitute.org

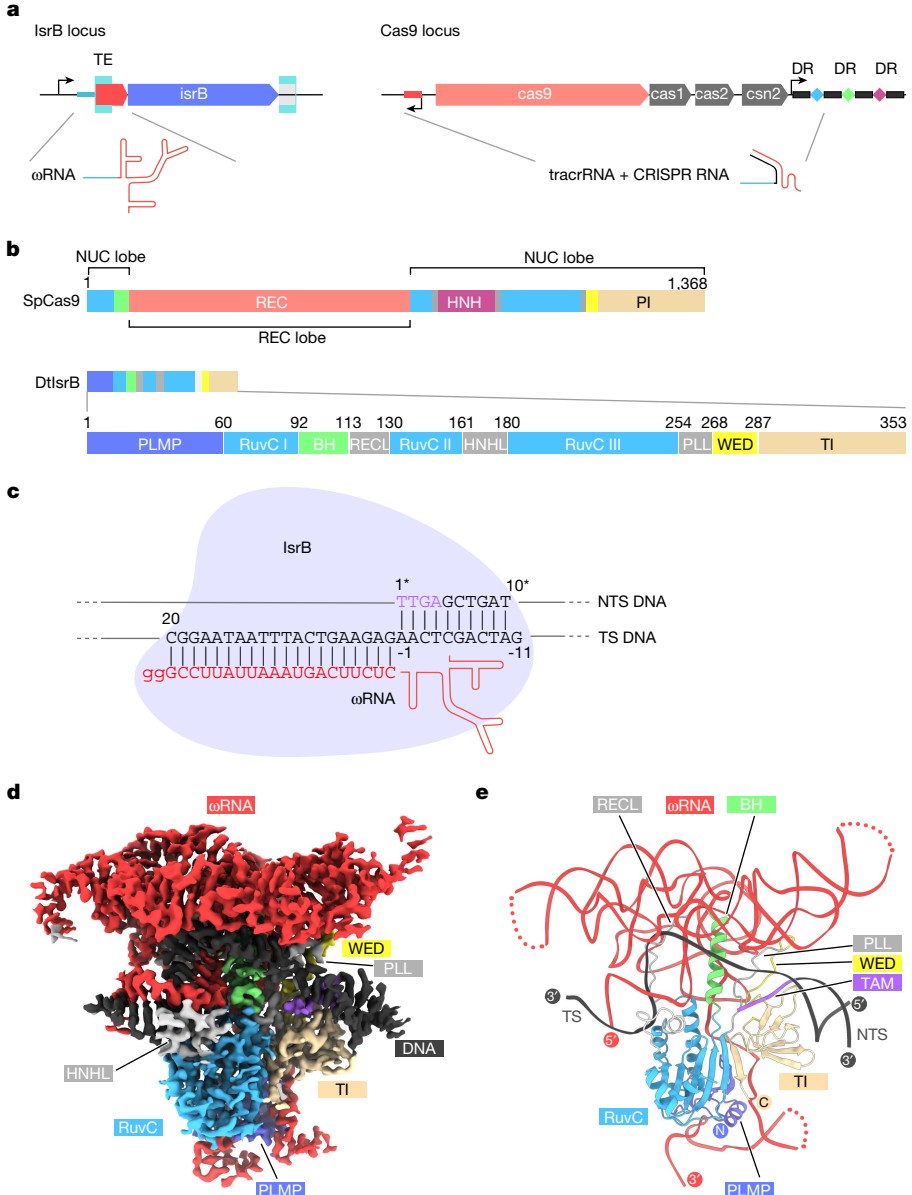

**Fig. 1 | Cryogenic-electron microscopy (cryo-EM) structure of the IsrB–ωRNA-target DNA complex. a**, Locus architecture and guide RNAs for IsrB (left) and Cas9 (right). **b**, Domain architecture of *Streptococcus pyogenes* SpCas9 (top) and *D. thermocuniculi* IsrB (DtIsrB) (bottom). **c**, Schematic of IsrB in complex with the ωRNA and the target DNA. The partial DNA duplex containing the TAM and target sequences used for the structural study are shown in sequence letters. **d,e**, Cryo-EM-density map (**d**) and structural model (**e**) of the IsrB–ωRNA-target DNA complex. Dashed lines represent poorly resolved regions of ωRNA. TE, transposon end; DR, direct repeat; NUC, nuclease; PI, PAM-interacting; PLL, phosphate-lock loop; TI, TAM-interacting; TS, target strand; NTS, non-target strand.

comprising two distorted β sheets (β1/2/6 and β3/4/5) and binds to the TAM-containing DNA duplex (Fig. 1e and Extended Data Fig. 3a). We denote this domain as the TAM-interacting (TI) domain because of structural and functional similarities to the PAM-interacting domain of Cas9 (Extended Data Fig. 3b). The extra β strand (β7) extensively interacts with the core fold of the TI domain and shares a common β sheet with the RuvC core that adopts the RNaseH fold (Extended Data Fig. 3a). This arrangement suggests that the TI and RuvC domains cooperate to define the distance between the RuvC active site and the TAM-binding site (Fig. 1e). The intermediate regions A (residues 254–267) and B (268–286) between the RuvC and TI domains seem to be functionally analogous to the phosphate-lock loop and WED domain of Cas9, respectively, and we therefore adopted those terms for IsrB (Fig. 1e). The PLMP domain (residues 1–59) features a four-stranded, antiparallel

β sheet (β1–4) and an α helix, and is structurally similar to the N-terminal domain of translation initiation factor 3 (Fig. 1e and Extended Data Figs. 3a and 4). In this domain, the PLMP motif-containing strand (β2) is bulged due to two prolines (Pro17 and Pro20) disrupting one of the hydrogen bonds, but seems to keep the integrity of a coherent strand (β1). The PLMP domain extensively interacts with the RuvC and TI domains, suggesting a role in supporting their functions.

## ωRNA architecture

The ωRNA consists of the 20-nt guide segment, which base pairs with the target DNA, and the 262-nt ωRNA scaffold. This scaffold consists of 12 helices (four stems (S1–4) and eight stem loops (SL1–8)), which are located on three layers (layer 1, S1/3 and SL1/2/5/6; layer 2, S2/4

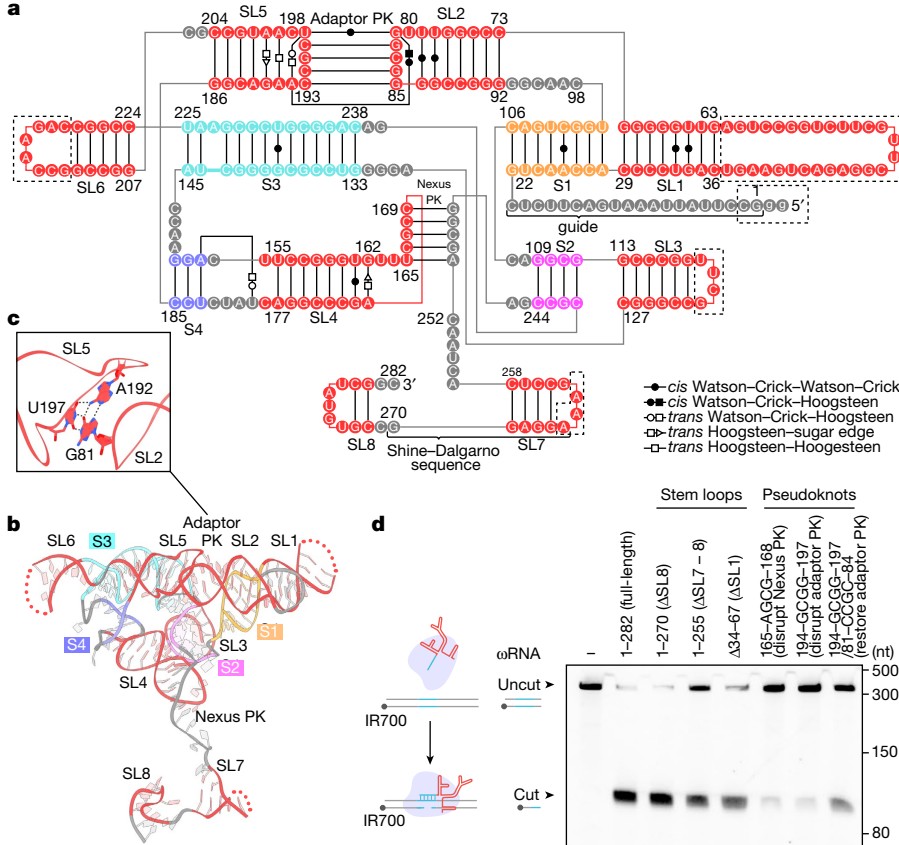

**Fig. 2 | Model of the DtIrsB ωRNA structure. a,b,** Schematic (**a**) and structural model (**b**) of the ωRNA scaffold (residues 21–282). S1–4, stem 1–4; SL1–8, stem loop 1–8; PK, pseudoknot. In **a**, canonical and non-canonical base pairs are depicted by solid black lines. Poorly resolved regions are enclosed in a dashed box. In **b**, the guide segment is omitted for clarity. **c,** A base-triple formation in the adaptor pseudoknot. Hydrogen bonds are shown as dashed lines. **d,** In vitro reconstituted DtIsrB-ωRNA RNP nicking of dsDNA substrates (with TTGA TAM)

with full-length ωRNA or truncated ωRNA. $n = 3$ independent technical replicates. Δ34–67, ωRNA in which nucleotides 34–67 were replaced with GAAA; 165-AGCG-168, ωRNA in which nucleotides 165–168 were replaced with AGCG; 194-GCGG-197, ωRNA in which nucleotides 194–197 were replaced with GCGG; 194-GCGG-197/81-CCGC-84, ωRNA in which nucleotides 81–84 and 194–197 were replaced with CCGC and GCGG, respectively.

and SL3/4; layer3, SL7/8) (Fig. 2a,b). All the RNA helices are packed together by various RNA interactions. The S1-SL1, S2-SL3 and S3-SL6 combinations are directly stacked in each combination. S4 and SL4 are co-axially stacked due to the direct stack between A152 and U154 and the base-triple formation among A152, U179 and U183. SL2 and SL5 form a pseudoknot (which we denote as the adaptor pseudoknot), which is capped by a base-triple formed by G81, A192 and U197 (Fig. 2c). Some RNA helices connect layers within the globular ωRNA structure. S2, C107, A108, G245 and A246 form the nexus region, which is widely conserved in the tracrRNA of Cas9s (ref. [9]) (Fig. 2a). This nexus region and S4 are directly connected to S1 and SL5, respectively, between layers 1 and 2. SL4 forms a pseudoknot (which we denote as the nexus pseudoknot) with the region between S2 and SL7, enabling interactions between layers 2 and 3 (Fig. 2a,b). Mutations disrupting base pairs in the pseudoknots abolished the DNA nicking activity, and subsequent mutations restoring base pairs in the adaptor pseudoknot partially restored this activity, highlighting the importance of the pseudoknots for ωRNA function (Fig. 2d). These structural and biochemical data show that the ωRNA forms a compact, globular structure achieved by various RNA interactions. Such a structure may be beneficial for OMEGA systems: if the ωRNA autonomously forms its globular structure and functions as a scaffold (in contrast to tracrRNA), the effector protein does not need auxiliary motifs/domains to support RNA folding and function. Furthermore, if the globular shape provides some resistance to endogenous RNA degradation, it could facilitate ωRNA functioning in *trans* with an effector protein. This latter possibility is supported by

the finding of standalone ωRNAs that can function with the related OMEGA effector IscB[4].

The 5′-stem region of ωRNA (S1, SL1 and SL2) is designated the guide adaptor region. It seems that during the evolutionary transition from OMEGA system to CRISPR–Cas, SL2 and the descending strands of S1/ SL1 of the ωRNA were adapted to form the CRISPR array to enable the formation of the functional Cas9–CRISPR RNA (crRNA)–tracrRNA complex (Fig. 1a). The genomic sequence encoding the guide adaptor region is important for IS200/IS605 transposon activity in bacterial genomes[10] (Fig. 2a). We truncated part of this region, SL1 (ΔSL1 ωRNA), and found that the resulting RNA still supported robust DNA nicking activity by IsrB (Fig. 2d). Furthermore, we reconstituted ΔSL1 ωRNA with the IsrB protein and target DNA and performed a single-particle analysis, generating a 6.9-Å resolution map (Extended Data Fig. 6a–e). Comparing this map with that of the full-length RNA validated the SL1 position determined from our RNA model and revealed conformational similarity between the full-length and ΔSL1 RNAs (Extended Data Fig. 6a,b). These results indicate that SL1 in the guide adaptor region is not required for target DNA nicking by IsrB and instead may contribute to other functions involved in the mobility of IsrB-encoding transposons. The ωRNA scaffold extensively interacts with all parts of IsrB except for the HNHL region (Fig. 1e). In particular, the PLMP domain interacts with the tandem hairpins (SL7 and SL8) near the 3′ end of the ωRNA. The truncation of SL7/8, but not SL8 reduced the nicking activity of IsrB (Fig. 2d). Given that the terminal hairpin (SL7) of the ωRNA contains the Shine–Dalgarno sequence located immediately upstream

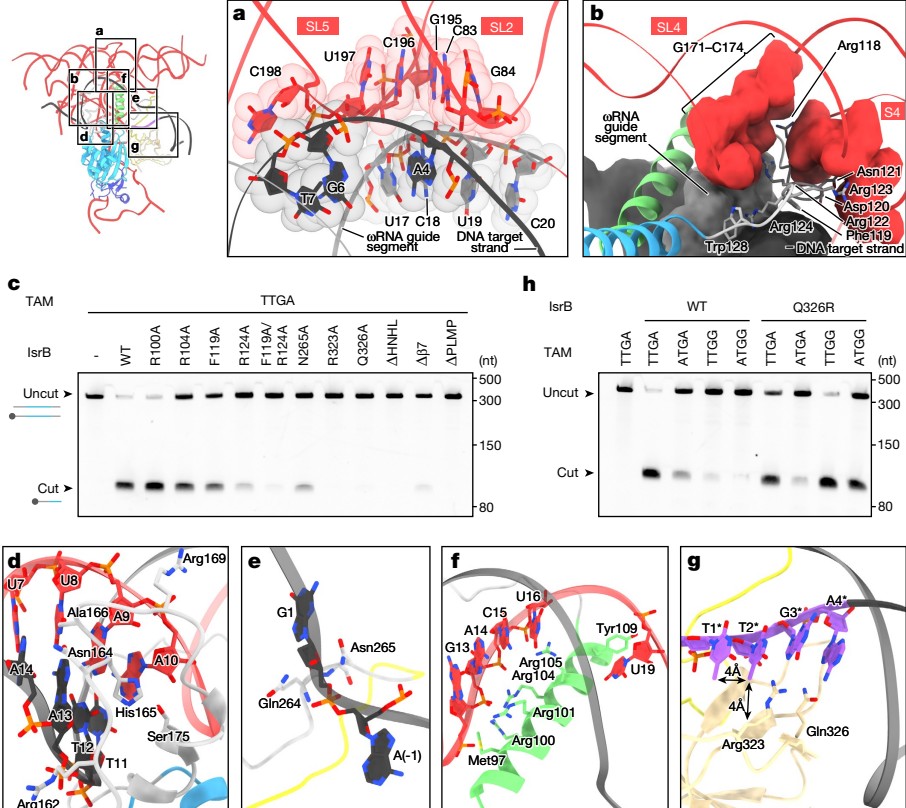

**Fig. 3 | DNA targeting and nicking mechanism of IsrB.** Inset shows the location of zoomed in panels. **a**, Heteroduplex recognition by the adaptor pseudoknot. **b**, Heteroduplex recognition by SL4, S4 and RECL. The volumes of RNA and DNA are generated from atomic coordinates, using Chimera X. **c**, In vitro reconstituted DtIsrB-ωRNA RNP nicking of dsDNA substrates (with TTGA TAM) with wild-type (WT) or mutant DtIsrB. $n = 3$ independent technical replicates. ΔHNHL, IsrB mutant in which residues 161–174 were replaced with a GSG-linker. Δβ7, IsrB mutant in which residues 341–353 were deleted. ΔPLMP, IsrB mutant in which residues 1–52 were deleted. To confirm the protein

stability of deletion mutants, we checked the protein expression in bacterial lysate overexpressing the deletion mutants (Extended Data Fig. 5b). **d**, Heteroduplex recognition by HNHL. **e**, Recognition of the +1 phosphate (phosphodiester bond between nucleotides dG1 and dA(−1) of target strand DNA) by the phosphate-lock loop. **f**, Recognition of the guide segment by BH. **g**, TAM recognition by the TI domain. **h**, TAM specificity of DtIsrB. In vitro reconstituted DtIsrB-ωRNA RNP nicking of dsDNA substrates (with TTGA/ATGA/TTGG/ATGG TAMs) with WT or mutant DtIsrB. $n = 3$ independent technical replicates.

of the IsrB-coding region, these results indicate that the IsrB–ωRNA interaction is important for IsrB function and could contribute to the regulation of IsrB expression in its native context.

## DNA-targeting mechanism of IsrB–ωRNA complex

We next sought to leverage structural information to decipher the DNA-targeting mechanism of IsrB. The gRNA–target DNA heteroduplex is surrounded by S2/S3/S4/SL2/SL4/SL5 of the ωRNA as well as the RuvC domain and the BH/RECL/HNHL regions of IsrB (Figs. 1e and 2b). SL2, SL4 and SL5 directly contact the heteroduplex backbone through hydrogen bonds and van der Waals interactions (Fig. 3a,b). S2, S3 and S4 indirectly recognize the heteroduplex backbone, using a short peptide linker, RECL, in which residues 113–124 are induced to fit into the grooves of S2/S3/S4 and the heteroduplex (Fig. 3b)[11]. Mutating F119 and R124 to alanine reduced the DNA nicking activity of IsrB, highlighting the functional importance of these residues in the RECL (Fig. 3c). In addition to the ωRNA, the IsrB protein binds extensively to the heteroduplex (Fig. 1e). The HNHL recognizes the minor groove of the heteroduplex through interactions with the backbone ribose moieties (Fig. 3d). We confirmed the importance of this interaction by deleting residues V161–F174 in the HNHL, which abolished the DNA nicking activity (Fig. 3c and Extended Data Fig. 5b). Several arginine residues in the BH contact the phosphate backbone of the ωRNA guide segment in a similar manner to that in the Cas9–guide RNA complex, in which

the guide RNA–BH interactions preorder the guide region for DNA recognition and unwinding[12] (Fig. 3f). Mutating R104, but not R100, to alanine reduced the DNA nicking activity of IsrB, highlighting the functional importance of R104 in the BH (Fig. 3c). Downstream of the target region (dG1–dC20), the ωRNA-complementary DNA strand (that is, the target strand) flipped and base-paired with the non-target DNA strand to form a TAM-containing duplex (dA[−1]-dA[−10]-dT1*-dT10*) (Fig. 1c,e). The backbone phosphate group between dC20 and dA(−1) in the target strand is recognized by Asn265 in the phosphate-lock loop, thereby facilitating heteroduplex formation (Fig. 3e). Mutating N265 to alanine reduced the nicking activity, suggesting the importance of this residue for DNA unwinding (Fig. 3c). The PLMP domain and the β7 motif in the TI domain are the pivotal units in the RuvC–TI–PLMP scaffold (Extended Data Fig. 3a). Truncating these domains/motifs abolished the DNA nicking activity of IsrB, indicating the importance of the rigid scaffold of RuvC–TI–PLMP (Fig. 3c and Extended Data Fig. 5b). These findings show that both IsrB and the ωRNA scaffold substantially contribute to the recognition of the guide–target heteroduplex for DNA targeting.

We previously found that DtIsrB shows a NTGA TAM preference[4], but given that DtIsrB is a thermophilic enzyme, we repeated the TAM identification assay at 60 °C. At this temperature, we observed a TTGA TAM preference (Fig. 4b). We then sought to characterize this preference structurally. The TAM-containing duplex is bound in the cleft between the WED and TI domains, in which the TAM-nucleobases in

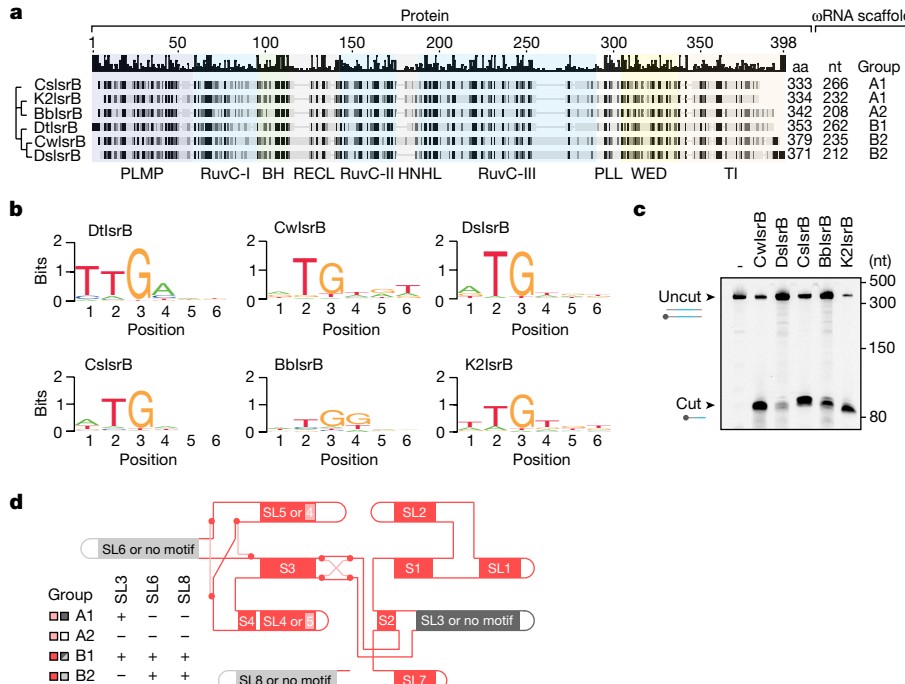

**Fig. 4 | IsrB diversity. a**, Phylogenetic tree of selected IsrB orthologues. Protein sizes are indicated, with domains highlighted in coloured boxes and conserved sequences in black. Cognate RNA sizes and groups (Fig. 4d) are indicated. **b**, TAM sequences for six IsrB orthologues using in vitro cleavage of a plasmid library containing randomized TAMs and the target sequence. **c**, In vitro reconstituted IsrB–ωRNA RNP nicking of dsDNA substrates with five IsrB orthologues. For CwIsrB, CsIsrB and K2IsrB, the target DNA contained a TTGA TAM. For DsIsrB and BbIsrB, the target DNA contained an ATGG TAM. *n* = 3 independent technical replicates. **d**, Structural models of the ωRNA scaffolds for six IsrB orthologues based on secondary structure predictions. The predicted ωRNA scaffolds are classified into groups A (subgroup A1, CsIsrB and K2IsrB; subgroup A2, BbIsrB) and B (subgroup B1, DtIsrB; subgroup B2,

CwIsrB and DsIsrB). In group A, SL2 and SL4 form pseudoknots, and SL5 and the intermediate region between S2 and SL7 form pseudoknots. Connecting regions that differ from group B are coloured pink. The intermediate region between SL5 and S3 as well as the terminal region after SL7 ('no motif', grey) are predicted to be unpaired nucleotides. In group B, SL2 and SL5 form pseudoknots, and SL4 and the intermediate region between S2 and SL7 form pseudoknots. Connecting regions (red) are as in group A. The intermediate region between SL5 and S3 as well as the terminal region after SL7 are predicted to be stem loops (SL6 and SL8, grey). In subgroups A1 and B1, the intermediate region between S2 and S3 is predicted to be a stem loop (SL3, dark grey), whereas in subgroups A2 and B2, that region is predicted to be unpaired nucleotides ('no motif', dark grey).

the non-target strand are read out by the residues in the TI domain (Figs. 1e and 3g). Although the dT1* nucleobase does not directly contact the protein, the C5 of the dT2* nucleobase forms van der Waals interactions with that of dT1* and the aliphatic portion of the Arg323 side chain, consistent with the preference for the first and second Ts in the TAM. The O6 and N7 of dG3* interact with R323, in line with the preference for the third G of the TAM. The R323A mutant lacked cleavage activity, supporting a role for R323 in TAM recognition (Fig. 3c). The N6 and N7 of dA4* interact with Gln326, consistent with the preference for the fourth A in the TAM. To test whether Q326 recognizes the fourth TAM nucleotide, we mutated this residue to alanine and found that this mutation abolished target cleavage (Fig. 3c). The wild-type IsrB showed cleavage activity on targets with TTGA/ATGA TAMs, but not with TTGG/ATGG TAMs (Fig. 3h). However, the Q326R mutant was active with all four of these TAMs. These results indicate that Q326 recognizes the fourth nucleotide in the TAM. In SpCas9, the PAM preference can be modified through alteration of the hydrogen-bonding interactions between the amino acid at position 1,335 (Arg in wild-type SpCas9 or Gln in SpCas9 VQR-variant) and the third nucleotide of the PAM (G or A, respectively)[13,14]. Analogously, in IsrB, the TAM preference can be modified through alteration of the hydrogen-bonding interactions between the amino acid at position 326 and the fourth nucleotide of the TAM. Together, these results indicate that DtIsrB recognizes the TTGA TAM in the non-target strand by a combination of hydrogen bonds and van der Waals interactions, and indicate that altering these interactions could expand the TAM preference.

To investigate the DNA nicking mechanism of IsrB, we identified the nicked site in the DNA by Sanger sequencing. IsrB nicked the non-target strand 8–11 nt upstream of the TAM (Extended Data Fig. 6a), in contrast to Cas9s, which cleave the non-target strand 2–5 nt upstream of the PAM[15]. To mimic the nicked product, we added 10 nt to the 5′ end of the non-target strand in the SL1-truncated IsrB complex structure (Extended Data Fig. 6b). We observed EM density of the extended part of the non-target strand, which is docked into the RuvC domain (Extended Data Fig. 6e). In the IsrB structures, the TAM and TAM-proximal parts of the non-target strand are removed from the RuvC domain (Extended Data Fig. 6e,f), whereas in the SpCas9 structure, the PAM-proximal part of the non-target strand interacts with the RuvC and HNH domains[16] (Extended Data Fig. 6g). The conformational difference between the non-target strands loaded onto the RuvC domains explains the distinct location of the DNA cut made by IsrB compared to that made by SpCas9.

## IsrB diversity

To assess the conservation of the ωRNA ternary structure across IsrBs, we identified five orthologues (CwIsrB, IsrB from *Crocosphaera watsonii*; DsIsrB, IsrB from *Dolichospermum* sp.; CsIsrB, IsrB from *Calditerricola satsumensis*; BbIsrB, IsrB from *Burkholderiales* bacterium; K2IsrB, IsrB discovered from contig k249_576930 of viral metagenome assembly) and their cognate ωRNAs (Fig. 4a). A TAM discovery assay showed that CwIsrB/K2IsrB/CsIsrB/DsIsrB recognize an NTG TAM, whereas BbIsrB recognizes an NTGG TAM (Fig. 4b). We confirmed the functionality of these ωRNAs and validated the TAM preferences using

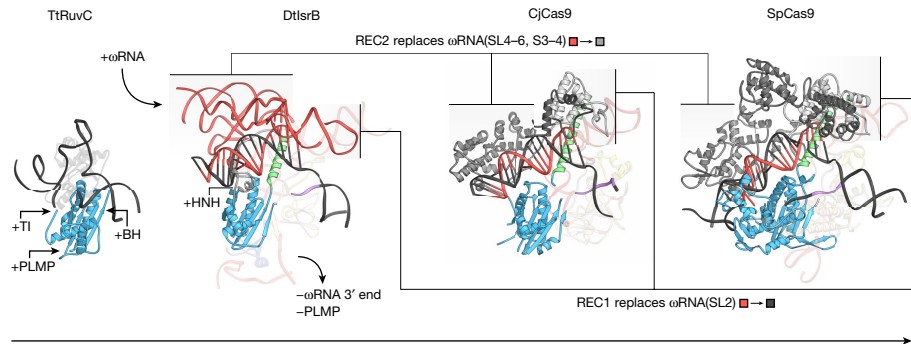

**Fig. 5 | Model of IsrB evolution.** Structural determinants of the evolution from ancestral RuvC nucleases to IsrB and then Cas9. Examples from modern descendants (extants) of each family are shown beginning with *T. thermophilus* RuvC (TtRuvC, PDB 6S16), DtIsrB, CjCas9 (PDB 5X2G) and SpCas9 (PDB 7S4X). Critical stages in the proposed evolutionary process are shown, including the insertions of the TI, PLMP and BH domains, interaction with ωRNA, insertion of the HNH domain, loss of the PLMP domain and replacement of various parts of the ωRNA with REC regions (domain replacements are shown with a colour key). The portion of REC2 in CjCas9 and SpCas9 that replace SL2 in the DtIsrB ωRNA are coloured in a dark grey. Connected base pairing is shown only for the guide–DNA duplex. Disconnected base pairing is shown for the ωRNA adaptor pseudoknot to highlight its position near the RNA–DNA duplex.

a DNA cleavage assay with the target DNA containing the single TAM (Fig. 4c). We generated 3D structure models of these IsrB orthologues and the covariance folded two-dimensional (2D) structure models of their cognate ωRNAs (Extended Data Fig. 7). The protein 3D-model and the RNA 2D model were compatible with the experimentally determined structures of DtIsrB and its cognate ωRNA, demonstrating the general reliability of structural prediction (Fig. 2a and Extended Data Fig. 7a,b). In the secondary structure prediction, the ωRNAs of DtIsrB and the other five orthologues maintain the core domain composition consisting of four stems (S1–4) and five stem loops (SL1/2/4/5/7) (Fig. 4d and Extended Data Fig. 7a). In the cryo-EM structure of the DtIsrB ωRNA (DtRNA), SL3, SL6 and SL8 are located at the periphery of the scaffold and do not contribute to the formation of the core (Fig. 2b). Truncation of SL8 did not appreciably affect DtIsrB cleavage activity, indicating that the ωRNAs lacking this motif support at least the minimal functionality of IsrB (Fig. 2d). In the ωRNAs of CwIsrB and DsIsrB, SL2 and SL5 as well as SL4 and the SL7-adjacent single-stranded region are predicted to form two pseudoknot structures, consistent with the structure of the DtRNA (Fig. 4d and Extended Data Fig. 7a). By contrast, in the ωRNAs of CsIsrB, K2IsrB and BbIsrB, two pseudoknot structures are predicted to be formed by SL2 and SL4 as well as SL5 and the SL7-adjacent single-stranded region (Fig. 4d and Extended Data Fig. 7a). This SL4–SL5 shuffling involved in the pseudoknot formation has been reported previously[4] and highlights the structural robustness of ωRNAs, which maintain overall similar structures despite structural rearrangements. Taken together, the demonstrated functionality of IsrB orthologues and the predicted structural similarities of IsrBs and their ωRNAs indicate the generality of the ωRNA-guided DNA-targeting mechanism suggested by the present cryo-EM structure.

## Discussion

To trace the protein domain evolution from IsrB to Cas9, we compared the structure of IsrB with the structure of one of the largest known IscBs (OgeuIscB)[17], a distant relative of IsrB containing the HNH nuclease domain, and the predicted structure of YnpsCas9-1 (an early branching Cas9 of subtype II-D from Ga0315277_10040887 that is among the Cas9s most closely similar to IscB)[4] (Extended Data Fig. 8). Apart from the gain of the HNH domain in IscB, we also observe big differences in other regions. For example, the RECL in some, but not all clades of IscB, is larger than the corresponding linker region in IsrB and folds into a minimal secondary structure, whereas in YnpsCas9-1, a large globular domain was acquired in the REC region. In other Cas9, such as SpCas9, this domain is even larger and more complex. The RuvC

domain in OgeuIscB contains a few larger loops, whereas in YnpsCas9-1, it contains long insertions that seem to have further evolved into highly structured domains in other Cas9s including SpCas9. This enlargement of the RuvC domain in Cas9 is accompanied by the loss of the PLMP domain. Similarly, the WED and TI domains have minimal size in other IsrBs and IscBs except specifically in OgeuIscB and other large IscBs in which these domains are expanded. The WED and TI domains probably continued expanding into the large, globular versions found in YnpsCas9-1 and SpCas9. SpCas9 harbours a larger PAM-interacting domain that contains an extra globular region located downstream of the common core PAM-interacting domain. The size reduction and split of the ωRNA into dual RNA guides in Cas9 (for example, tracrRNA–crRNA) probably accompanied the acquisition of the REC domain and the overall enlargement of all domains of Cas9.

To characterize in greater detail the minimization of the ωRNA as it evolved into cr/tracrRNAs, we compared the structure of DtIsrB ωRNA (DtRNA) with those of OgeuIscB ωRNA (OgRNA), CjCas9 single-guide RNA (CjRNA) and SpCas9 sgRNA in their protein/target DNA-bound states (Extended Data Fig. 9)[16–18]. On the basis of topology, location and secondary structure, we mapped DtRNA structural features (S1–4 and SL1–8) on other RNA species and named unidentified structural motifs as motifs 1–5 (M1–5). The structures of the 5′-stem region (S1 and SL1 in DtRNA) and the nexus region (S2 in DtRNA) are conserved in all four RNA species. The ascending strand of the 5′-stem region is replaced with crRNA in the evolutionary transition from OMEGA-IsrB/IscB to CRISPR–Cas9. Moreover, as ωRNAs evolved into tracrRNAs, the inserted helices (S3/S4/SL4/SL5/SL6 in DtRNA) within the nexus region degenerated, contributing to the compaction and simplification of the RNA structure. The SL4 motifs of DtRNA and OgRNA form nexus pseudoknots that are conserved in ωRNAs, whereas some base pairings in CjRNA M3 are well superposed with those nexus pseudoknots. An embedded stem loop in DtRNA 5′-stem region (SL2) base pairs with one of the embedded stem loops in the nexus region (SL5), forming a functional pseudoknot (adaptor pseudoknot) that recognizes the target DNA. One base adjacent to the adaptor pseudoknot (C198), forms several contacts between 3 and 5 Å with the phosphate and deoxyribose moieties of the DNA at position 6 (G6) and 7 (T7) (Fig. 3a), conferring a unique adaptation in which the ωRNA scaffold can recognize the RNA–DNA duplex. The adaptor pseudoknot is conserved in IsrB ωRNAs but is degenerated in the transition to IscB ωRNAs and Cas9 tracrRNAs, a change that correlates with and is probably compensated by the REC-region expansion.

We also sought to better understand the mechanistic changes associated with the domain acquisitions in IsrB and Cas9 during their

evolution from the compact RuvC-like ancestor. To this end, we compared the target-bound structures of *Thermus thermophilus* RuvC (TtRuvC), IsrB, CjCas9 and SpCas9 (Fig. 5). As RuvC domain-containing proteins evolved to interact with ωRNAs, they acquired TI/PI, PLMP and BH domains. In the structures of both IsrB and Cas9, the RuvC, WED, TI/PI and BH domains as well as the phosphate-lock loop form a functional core with similar configurations; the guide–target heteroduplex and the TAM/PAM duplex are bound to this core in a similar position and orientation. The TI/PI domain recognizes the TAM/PAM nucleobases, probably functioning as a primer for target DNA unwinding and heteroduplex formation, with the assistance of the phosphate-lock loop, BH and ωRNA/gRNA. Although IsrB and Cas9 share homologous RuvC and BH domains, IsrB (as well as IscB) uniquely contains the PLMP domain, which directly interacts with RuvC I. Examination of the IsrB structure further reveals a role of the PLMP domain in stabilizing the base of the terminal hairpin of the ωRNA and contacting the Shine–Dalgarno sequence. Furthermore, IsrB contains only minimal RECL and HNHL regions (17 and 19 amino acids, respectively, in DtIsrB), and they probably play different roles in DNA targeting from those performed by the larger REC lobe and HNH domain in Cas9 (for example, 625 and 135 amino acids, respectively, in SpCas9). In SpCas9, the REC lobe probes the target DNA through interactions with the heteroduplex, activates the DNA-bound RuvC nuclease through the communication with the HNH domain and facilitates R-loop formation[19–21]. However, in IsrB, this interdomain communication is probably aided by the ωRNA both through backbone-backbone and base-backbone interactions because RECL and HNHL are comparatively small.

The comparatively large ωRNA (roughly 300-nt compared to 100-nt sgRNA used by Cas9) seems to contribute to the connection between DNA targeting and nicking activities, compensating for the small RECL and HNHL regions (Extended Data Fig. 10). In the multi-layered ωRNA architecture, the upper layer RNA helices (S2/S3/S4/SL2/SL4/SL5), which form an interaction network for ωRNA-driven heteroduplex recognition, are associated with the lower layer RNA helices (SL7/SL8) and extensively interact with the nicking module (PLMP/RuvC/TI domains) by the nexus pseudoknot interactions between S2, SL4 and SL7. Given that mutations in the adaptor pseudoknot in the ωRNA abolished the nicking activity of IsrB (Fig. 2d), even though the pseudoknot is distant from the target DNA, the ωRNA structural motifs could be important for allosteric regulation of DNA sensing by the ωRNA/RECL and DNA nicking by the RuvC nuclease domain, providing an avenue for integrating further forms of regulation. This ωRNA-driven allosteric regulation mechanism is supported by the overall high surface charge and area through which IsrB contacts ωRNA. Other large (roughly 400–900-nt) functional non-coding RNAs, such as group I intron, group II intron and Ribonuclease P, have complex ternary structures and their peripheral regions can control their central catalytic cores by allosteric mechanisms[22–25]. Future structural studies of IsrB in other conformations, such as the catalytically active IsrB R-loop complex, will address this hypothesis and deepen our mechanistic understanding of OMEGA systems.

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

## Methods

### Electron microscopy sample preparation

The gene encoding full-length DtIsrB (residues 1–353) was codon optimized, synthesized (Twist Bioscience) and cloned into a modified pC013 vector (Addgene Plasmid no. 90097). The DtIsrB-coding region consists of His$_6$-Twinstrep-tag, SUMO-tag, DtIsrB and GFP-tag. Wild-type DtIsrB was expressed at 18 °C in *Escherichia coli* Rosetta(DE3) pLysS cells (Novagen). *E. coli* was cultured at 37 °C in Luria-Bertani medium (containing 100 mg l$^{-1}$ ampicillin) until the OD$_{600}$ reached 0.5, and then protein expression was induced by the addition of 0.1 mM isopropyl-β-ᴅ-thiogalactopyranoside and incubation at 18 °C for 20 h. The *E. coli* cells were resuspended in buffer A (50 mM Tris-HCl, pH 8.0, 20 mM imidazole and 1 M NaCl), lysed by sonication and then centrifuged. The supernatant was mixed with Ni-NTA Agarose (Qiagen). The protein-bound column was washed with buffer A, buffer B (50 mM Tris-HCl, pH 8.0, 20 mM imidazole and 0.3 M NaCl) and buffer C (50 mM Tris-HCl, pH 8.0, 0.3 M imidazole and 0.3 M NaCl). The protein was eluted with buffer D (50 mM Tris-HCl, pH 8.0, 0.3 M imidazole and 1 M NaCl). The cognate ωRNA of DtIsrB was transcribed in vitro with T7 RNA polymerase, using a PCR-amplified DNA template and HiScribe T7 Quick High Yield RNA Synthesis kit (NEB). The template consists of the T7 promoter (TAATACGACTCACTATAGG), guide (GCCTTATTAAATGACTTCTC) (residues 1–20) and ωRNA scaffold (residues 21–282). The transcribed RNA was purified using an RNeasy kit (Qiagen) according to the manufacturer's instructions. The target and non-target DNA strands (GATCAGCTCAAGAGAAGTCATTTAATAAGGC and TTGAGCTGAT, respectively) were purchased from GENEWIZ. For the reconstitution of complex A, the purified DtIsrB protein was mixed with the ωRNA, the target DNA strand and the non-target DNA strand (the TTGA TAM) (molar ratio, 2.3:1:7:7) in buffer E (10 mM Tris-HCl, pH 8.0 and 50 mM NaCl, 5 mM MgCl$_2$) and incubated at 37 °C for 15 min. Complex A was purified by gel filtration chromatography on a Superose 6 Increase 10/300 column (Cytiva) equilibrated with buffer F (20 mM HEPES-NaOH, pH 7.0 and 50 mM NaCl, 5 mM MgCl$_2$). Complex A (final concentration: 0.1 mg ml$^{-1}$) was incubated with BS3 (final concentration: 0.5 mM) at 4 °C for 2 h. For the reconstitution of complex B, the lambda N protein (MDAQTRRRERRAEKQAQWKAAN) was inserted between DtIsrB and GFP-tag. Residues 34–67 of ωRNA scaffold (residues 21–282) were replaced by a GAAA linker. The GAAA linker-fused boxB RNA (GAAAGCCCUGAAGAAGGGC) (residues 283–302) was appended to the 3′ end of the ωRNA scaffold. The same target DNA strand was used for this reconstitution. The 5′ extended non-target DNA strand (TACTGAAGAGTTGAGCTGAT) was purchased from GENEWIZ. The purified DtIsrB protein was mixed with the ωRNA, the target DNA strand, and the non-target DNA strand (the TTGA TAM) (molar ratio, 2.3:1:1.5:1.5) in buffer G (10 mM Tris-HCl, pH 8.0 and 50 mM NaCl) and incubated at 37 °C for 15 min. Complex B was purified by the same size-exclusion column equilibrated with buffer G. For the grid preparation, purified complex A and B solutions (0.1 mg ml$^{-1}$, 3 µl) were applied to freshly glow-discharged UltrAuFoil 300 mesh R1.2/1.3 grids (Quantifoil) in a Vitrobot Mark IV (FEI) at 4 °C with a waiting time of 0 and 10 s and a blotting time of 2 and 4 s under 95% humidity, respectively.

### Electron microscopy data collection and processing

Cryo-EM data for complex A were collected at HHMI Janelia Research Campus using a Titan Krios G2 microscope (Thermo), operated at 300 kV and equipped with a Gatan Bioquantum energy filer (Gatan) and a postfilter K3 direct electron detector (Gatan) in the electron counting mode. Each video was recorded at a nominal magnification of ×105,000, corresponding to a 0.839 Å per physical pixel (0.4195 Å per super-resolution pixel) at the electron exposure of 12.075 electrons per Å$^2$ per second and total exposure time was 5.0 s, resulting in an accumulated exposure of 60 e$^-$/Å$^2$. Then 50 frames per video were collected at 1.2 e$^-$/Å$^2$ dose per frame for a total of 60 e$^-$/Å$^2$ dose per video. The nominal defocus range was set at −0.8 to −2.2 µm. Automated data collection was carried out using scripts in SerialEM. For each stage position, image shift was used to collect data from nine holes with two video acquisitions per hole. Image shift induced beam tilt was calibrated and beam-tilt correction was applied during the data collection. Cryo-EM data for complex B were collected at MIT.nano using a Talos Arctica G2 microscope (FEI), operated at 200 kV and equipped with a Falcon 3EC direct electron detector (Thermo) in the linear mode. Each video was recorded at a nominal magnification of ×120,000, corresponding to a calibrated pixel size of 1.2550 Å at the electron exposure of 24.54 e$^-$/pix s$^{-1}$ for 3.99 s, resulting in an accumulated exposure of 62.53 e$^-$/Å$^2$. Next, 20 frames per video were collected at 3.1265 e$^-$/Å$^2$ dose per frame for a total of 62.53 e$^-$/Å$^2$ dose per video. The nominal defocus range was set at −2.6 to −1.0 µm. Automated data collection was carried out using the EPU software (Thermo). For each stage position, image shift was used to collect data from nine holes. To obtain the 3D reconstruction of complex A, data were processed using RELION-4.0 (ref. [26]). The video frames were aligned in 5 × 5 patches and dose weighted in Motion-Cor2 (ref. [27]). Defocus parameters were estimated by CTFFIND-4.1 (ref. [28]). From the 4,142 preprocessed micrographs, 1,626,574 particles were picked up by TOPAZ based auto-picking[29] and extracted in 3.146 Å pixel$^{-1}$. The selected 107,066 particles were then re-extracted in 1.144 Å pixel$^{-1}$ and subjected to one round of 3D refinement and 3D classification without alignment. The selected 58,188 particles were subjected to per-particle defocus estimation and Bayesian polishing. For beam-tilt refinement, the optics group of each micrograph is set on the basis of their hole position from stage. The polished particles were subjected to 3D refinement, and yielded a map with a global resolution of 3.10 Å according to the Fourier shell correlation 0.143 criterion. To obtain the 3D reconstruction of complex B, data were processed using the same programs. From the 2,542 motion-corrected and dose-weighted micrographs, 1,595,800 particles were picked up by TOPAZ based auto-picking and extracted in 3.138 Å pixel$^{-1}$. These particles were subjected to several rounds of 2D and 3D classifications. The selected 50,661 particles were then re-extracted in 1.255 Å pixel$^{-1}$ and subjected to homogeneous refinement using cryoSPARC[30], yielding a map with a global resolution of 6.85 Å according to the Fourier shell correlation 0.143 criterion.

### Model building and validation

The initial protein model was generated using AlphaFold2 (ref. [31]) under the ColabFold framework using default parameters and MMseqs2 to search for homologues into the ColabFold database[32], and manually modified using COOT[33] and ISOLDE[7] against the density map of complex A. The initial nucleic acid model was built with auto-DRRAFTER using the density map of complex A and the covariance-based secondary structure model of ωRNA[8]. The ωRNA (query) secondary structures were predicted using cmsearch[34] with the −max option to identify the highest scoring IscB/IsrB ωRNA covariance model from a previous study[4]. For the best model, query regions aligning to the model were assigned secondary structures from the model's predictions. Stem loop secondary structures that were found to be erroneously assigned to base pairs with one of the base identities equalling a gap character were reassigned to having no secondary structure. Secondary structures for query regions without coverage (≥8 bp of no match to the best covariance model), barring the low conservation region at the 3′ end beyond the nexus, were then predicted using mfold[35]. Pseudoknots were assigned manually by identifying matching base pairs at the pseudoknot locations expected for the given ωRNA type. ωRNA coordinates were modelled with auto-DRRAFTER starting from a slightly modified version of the covariance-based secondary structure model in which all non-canonical base pairs and most helices consisting of just a single base pair were removed. The dot-bracket notation for this secondary structure is provided below:

```
.((((((((((((((((((((((((((.((.((((((((((...((((((((((....))))))))...)))))))).
(((((((((({..}))))))))........)).)))).((.((((((....))))))....(((((((((((.....(((..(((((((...
[[[[[.))))))).....)))((((...}..}....)))).((.((...(.....)).)))..))))))))))))...))..]]]]]......
(((((....)))).(((.....))))..<<<<<<<<<<))))))))))))))))))))))>>>>>>>>>
```

All DNA nucleotides were modelled as RNA because auto-DRRAFTER cannot model DNA nucleotides. The guide/ωRNA scaffold/target DNA/non-target DNA were assigned to residues 1–20/21–282/283–313/314–323, respectively. The full RNA sequence used for modelling is provided below:

ggccuuauuaaaugacuucucgucaaccaccccugacugaagucagaggcuugcuu
cuggccugaguuggggggcccgguuuggcggggccgggggcaacuggcugaccaggc
ggcccgguucgccgggcagggguccgcggggcuaccaaggacuuccgggguguuucg
ccagcccggacuaucuccggcagaaccgcucaaugccgcggccggccaagaccggccu
aagcccugcggacagcgccgaggcgacaaucacuccgaaaggaggccguguaucggc
gaucagcucaagagaagucauuuaauaaggcuugagcugau

Auto-DRRAFTER modelling was performed in the absence of protein coordinates using the density map with regions corresponding to protein density removed. All initial rounds of modelling were performed in a preliminary 4.3-Å resolution density map. The modelling was set up manually by fitting helices corresponding to residues W:1–14 W:41–48 W:53–60 W:258–269 W:271–281 X:2–11 X:18-31 Y:1-10 into the density map. In the second round of auto-DRRAFTER modelling, the helix corresponding to residues W:41–48 and W:53–60 was allowed to move from its initial placement. Five rounds of modelling were performed, followed by one final round of modelling. For each round, between 2,000 and 6,000 models were built. One of the top ten scoring models was selected for further refinement by ISOLDE and Phenix[6], together with the protein model, to optimize the geometry and improve the fit to the cryo-EM density. After inspecting the optimized model and covariance-based secondary structure, two more rounds of auto-DRRAFTER modelling, including one final round, were performed in which the base pairing for the adaptor pseudoknot was modified slightly so that residues 81–84 and 194–197 were paired rather than residues 81–84 and 193–196. For this extra modelling, only residues W:73–99 and W:186–206 were rebuilt; all other residues remained fixed. One more final round of modelling was performed using the 3.1 Å resolution density map low-pass filtered to 4 Å. The final convergence of these models (pairwise root mean square deviation between models) is 4.1 Å. Auto-DRRAFTER convergence values have previously been shown to be predictive of model accuracy. Using a previously determined linear relationship between convergence and model accuracy (accuracy of 0.61 × convergence + 2.4 Å), the estimated accuracy of these initial computationally generated models is 4.9 Å. To further improve the accuracy, one of these models was refined with COOT, ISOLDE and Phenix together with the protein to produce the final IsrB–ωRNA-target DNA complex model. The final model (lacking protein residues 1–5/211–224/348–353, RNA residues 1–2/37–64/119–122/212–219/263–265 and target DNA residues 1/30–31, which were poorly resolved and omitted from the final model) was evaluated by MolProbity[36] and Q-score[37]. Molecular graphics and EM density figures were prepared with CueMol (http://www.cuemol.org), PyMOL (https://pymol.org/2/), UCSF Chimera (https://www.cgl.ucsf.edu/chimera/) or Chimera X (https://www.cgl.ucsf.edu/chimerax/).

### In vitro cleavage experiment
The IsrB protein and ωRNA templates were prepared for an in vitro transcription/translation expression system. The IsrB protein template consists of the T7 promoter and translation initiation sequences (GCGAATTAATACGACTCACTATAGGGCTTAAGTATAAGGAGG AAAAAAATATG), IsrB ORF sequence and T7 terminator sequence (CTAGCA TAACCCCTTGGGGCCTCTAAACGGGTCTTGAGGGGTTTTTTG). The ωRNA template consists of the T7 promoter sequence (GGAAATT AATACGACTCACTATAGG) and ωRNA sequence. The IsrB protein and ωRNA templates were embedded in the modified pC013 vector (Addgene Plasmid no. 90097) and the pCOLADuet-1 vector. Mutations

in the IsrB protein and ωRNA were introduced by a PCR-based method and the sequences were confirmed by DNA sequencing. The 320-bp PCR-amplicon (30 ng), which contains the 20-nt target sequence and the TAM and was fluorescently labelled by 5′ IRDye 700 (IDT), was incubated with the IsrB protein template (50 ng) and the ωRNA template (125 ng) in 12.5 μl of reaction buffer, containing 5 μl Solution A and 3.75 μl Solution B of PURExpress In vitro Protein Synthesis Kit (NEB). The reaction conditions were optimized as follows. Fig. 2d, 3 h: 2 h at 37 °C, 1 h at 60 °C; Fig. 3c, 2.1 h: 2 h at 37 °C, 5 min at 60 °C; Fig. 3h, 3 h: 2 h at 37 °C, 1 h at 60 °C; Fig. 4c (CwIsrB, DsIsrB and BbIsrB), 6 h at 37 °C; Fig. 4c (CsIsrB), 6 h: 2 h at 37 °C, 4 h at 60 °C; Fig. 4c (K2IsrB) and 2 h at 37 °C. DtIsrB is derived from a thermophilic organism, *D. thermocuniculi*, which grows at 60–80 °C (ref. [38]). The reaction was stopped by the addition of 3 μg of RNase A (Qiagen) and 0.24 units of Proteinase K (NEB). The reaction products were purified using a Wizard SV Gel and PCR Clean-Up System (Promega), resolved on a Novex 10% TBE-Urea Gel (Invitrogen) and then visualized using a ChemiDoc Imaging System (Bio-Rad). To examine the protein stability of deletion mutants, IsrB proteins were produced in the bacterial expression system used in the cryo-EM sample preparation. The *E. coli* cells were resuspended in buffer A (50 mM Tris-HCl, pH 8.0, 20 mM imidazole and 1 M NaCl), lysed by sonication and then centrifuged. The supernatant was mixed with MagneHis beads (Promega). The protein-bound column was washed with buffer A. The protein was eluted with buffer B (50 mM Tris-HCl, pH 8.0, 0.3 M imidazole and 1 M NaCl) and analysed by SDS–PAGE (Extended Data Fig. 5b). To determine the IsrB DNA cleavage sites, the 816-bp PCR-amplicon (400 ng) containing the 20-nt target sequence (GCCTTATTAACCTCAGCCTC) and the TAM was incubated with the IsrB protein template (100 ng) and the ωRNA template (125 ng) in 25 μl of reaction buffer, containing 10 μl Solution A and 7.5 μl Solution B of PURExpress In vitro Protein Synthesis Kit. After purifying the reaction product, the nicked product was cleaved using Nb.BbvCI (NEB). The cleaved products were gel-extracted, purified and analysed by DNA sequencing (GENEWIZ).

### IsrB and ωRNA curation and analysis
Representative IsrBs with intact RuvC active catalytic site residues and no signs of truncations were selected from among the three major clades of IsrBs as identified in a previous study[4], corresponding to IsrBs with ωRNAs of type G1b, G1c and G1h. ωRNAs corresponding to each IsrB were taken from the predictions in a previous study[4] and modified such that the end of the ωRNA occurred at the start of the IsrB. IsrBs were then discarded if the corresponding ωRNA's secondary structure, as determined by mfold, did not contain the conserved stem loops and pseudoknots (as manually identified) found in the covariance-based ωRNA secondary structure for the given ωRNA type[35]. The analysis nominated the CwIsrB, CsIsrB, DsIsrB, BbIsrB, K2IsrB sequences and corresponding ωRNAs. Covariance-based secondary structure and pseudoknot predictions were determined for the corresponding ωRNAs as described for the DtRNA. All ωRNAs were then visualized using forna[39].

For analysis of the PLMP domain, the DtIsrB PLMP domain was searched in HHPred for remote homologues, identifying IF-3 as a putative remote homologue. Representative sequences containing IF-3-N-terminal regions and PLMP domains from the IscB/IsrB family were obtained from UniProt and the National Center for Biotechnology Information, and aligned using MAFFT-einsi. Structural representatives were aligned and superimposed using the pymol super function.

### TAM identification
The TAM identification assay was performed using a TAM library, prepared as previously described[4]. Single-stranded DNA oligonucleotides (IDT), containing eight randomized nucleotides downstream of a 20-nt target sequence (GCCTTATTAACCTCAGCCTC), were converted to dsDNA by fill-in with PrimeSTAR Max DNA Polymerase (Takara) and

cloned into pUC19 by Gibson cloning (NEB) to generate a TAM library. The library (25 ng) was digested using an in vitro transcription/translation expression system containing the IsrB protein (50 ng) and ωRNA (125 ng) templates, as described in the in vitro cleavage experiment section. The reactions of CwIsrB, DsIsrB, CsIsrB, BbIsrB and K2IsrB were incubated for 4 h: 2 h at 37 °C, 1 h at 50 °C and 1 h at 60 °C. The reaction of DtIsrB was incubated for 3 h: 2 h at 37 °C and 1 h at 60 °C. It was then stopped by the addition of 3 μg of RNase A (Qiagen) and 0.24 units of Proteinase K (NEB). The reaction products were purified using a Wizard SV Gel and PCR Clean-Up System (Promega), and digested using Nb.BbvCI (NEB). The purified reaction products were subjected to end labelling and adaptor ligation using an NEBNext Ultra II End Repair/dA-Tailing Module (NEB), an NEBNext Ultra II DNA Library Prep Kit for Illumina (NEB) and an NEBNext Adaptor for Illumina (NEB). The USER Enzyme (NEB)-digested and purified DNA was amplified with a 12-cycle PCR using one primer specific to the TAM library backbone and one primer specific to the NEBNext adaptor, and with a subsequent 18-cycle PCR to add the Illumina i5 adaptor. To normalize the distribution of the 8N degenerate flanking sequences, the library plasmid was amplified with a 12-cycle PCR using primers specific to the library backbone and with a subsequent 18-cycle PCR to add the Illumina i5 adaptor. The amplified libraries were isolated on 2% agarose E-gels (Invitrogen) and sequenced on a MiSeq sequencer (Illumina). The resulting sequence data were analysed by extracting the six nucleotide TAM regions, counting the individual TAMs and normalizing the TAM to the total reads for each sample. Sequence motifs were generated using the selected TAMs in the top scoring fraction with the custom Python script used in our previous report[4].

## Reporting summary

Further information on research design is available in the Nature Research Reporting Summary linked to this article.

## Data availability

The atomic coordinates of the IsrB ternary structure have been deposited with the Protein Data Bank (PDB) at http://www.pdb.org (PDB 8DMB). The three-dimensional cryo-EM reconstructions of complex A and complex B have been deposited with the Electron Microscopy Data Bank (complex A EMD27533; complex B EMD26723).

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

**Acknowledgements** We thank E. Brignole, C. Borsa, X. Zhao and S. Yang for assistance with cryo-EM grid preparation and data collection. Specimens were prepared and imaged at the Cryogenic-Electron Microscopy Facility in MIT.nano, established in part with financial support from the Arnold and Mabel Beckman Foundation. We thank all members of the Zhang laboratory for helpful discussions and support. S.H. is supported by a JSPS Overseas Research Fellowship. K.K is supported by the Schmidt Science Fellows, in partnership with the Rhodes Trust, and the HHMI Hanna H. Gray Fellows Program. F.Z. is supported by National Institutes of Health (grant nos. 1DP1-HL141201 and 2R01HG009761-05); the Howard Hughes Medical Institute; the Poitras Center for Psychiatric Disorders Research at MIT; the Hock E. Tan and K. Lisa Yang Center for Autism Research at MIT; the Yang-Tan Molecular Therapeutics Center at McGovern, the BT Charitable Foundation and by the Phillips family and J. and P. Poitras.

**Author contributions** S.H. and F.Z. conceived the project. S.H., K.K., G.F., H.A.-T., M.E.W., S.K., F.E.D. and K.S.M. designed and performed experiments and analysed the results. R.Y., M.S. and Z.Y. performed data collection. F.Z. supervised the research and experimental design with support from R.K.M. S.H., R.K.M., E.V.K. and F.Z. wrote the manuscript with input from all authors.

**Competing interests** F.Z. is a scientific adviser and cofounder of Editas Medicine, Beam Therapeutics, Pairwise Plants, Arbor Biotechnologies and Proof Diagnostics.

**Additional information**
**Correspondence and requests for materials** should be addressed to Feng Zhang.

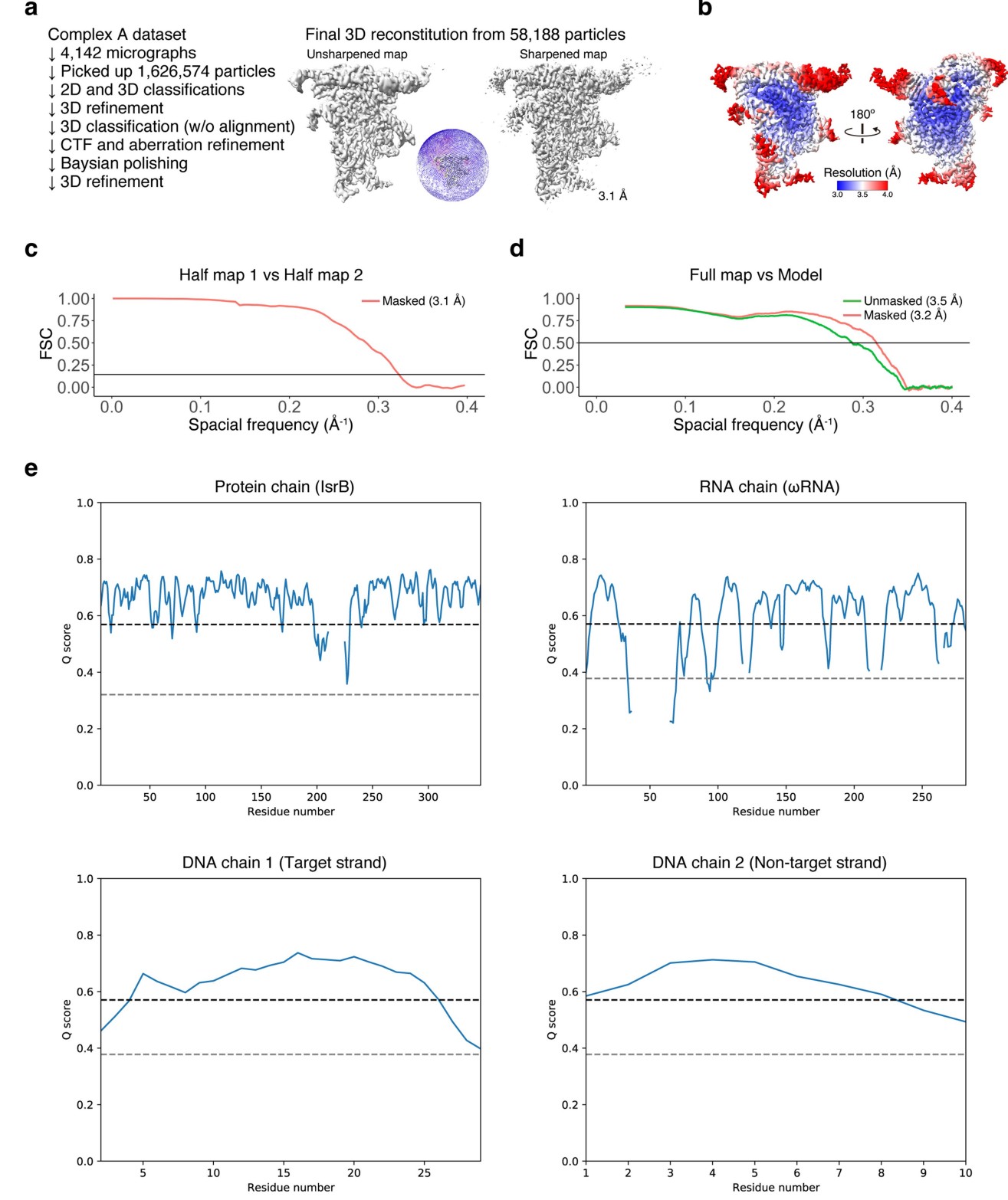

**Extended Data Fig. 1 | Cryo-EM data processing for the IsrB-ωRNA-DNA complex (complex A).** (a) Cryo-EM data processing schematic for single particle analysis of the complex A. Unsharpened (Left) and sharpened (Right) maps in the final 3D refinement. Particle orientation distribution (Center). (b) Final refined map, colored by local resolution, calculated in RELION-4.0 with FSC threshold 0.5.(c) FSC curves calculated between the half maps of complex A from the final round of the refinement in RELION-4.0. (d) FSC curves calculated between the model and the final refined map, using phenix. validation_cryoem. (e) Q-scores for each residue of IsrB-ωRNA-target strand DNA-non-target strand DNA model in 3.1 Å map of the IsrB-ωRNA-DNA complex. The dashed black and grey lines in the plot represent the expected Q-scores based on the global map resolution (3.1 Å) and the local map resolution (4.5 Å), respectively. Q-scores for the RNA and DNA residues are consistent with the expected values based on the local map resolution.

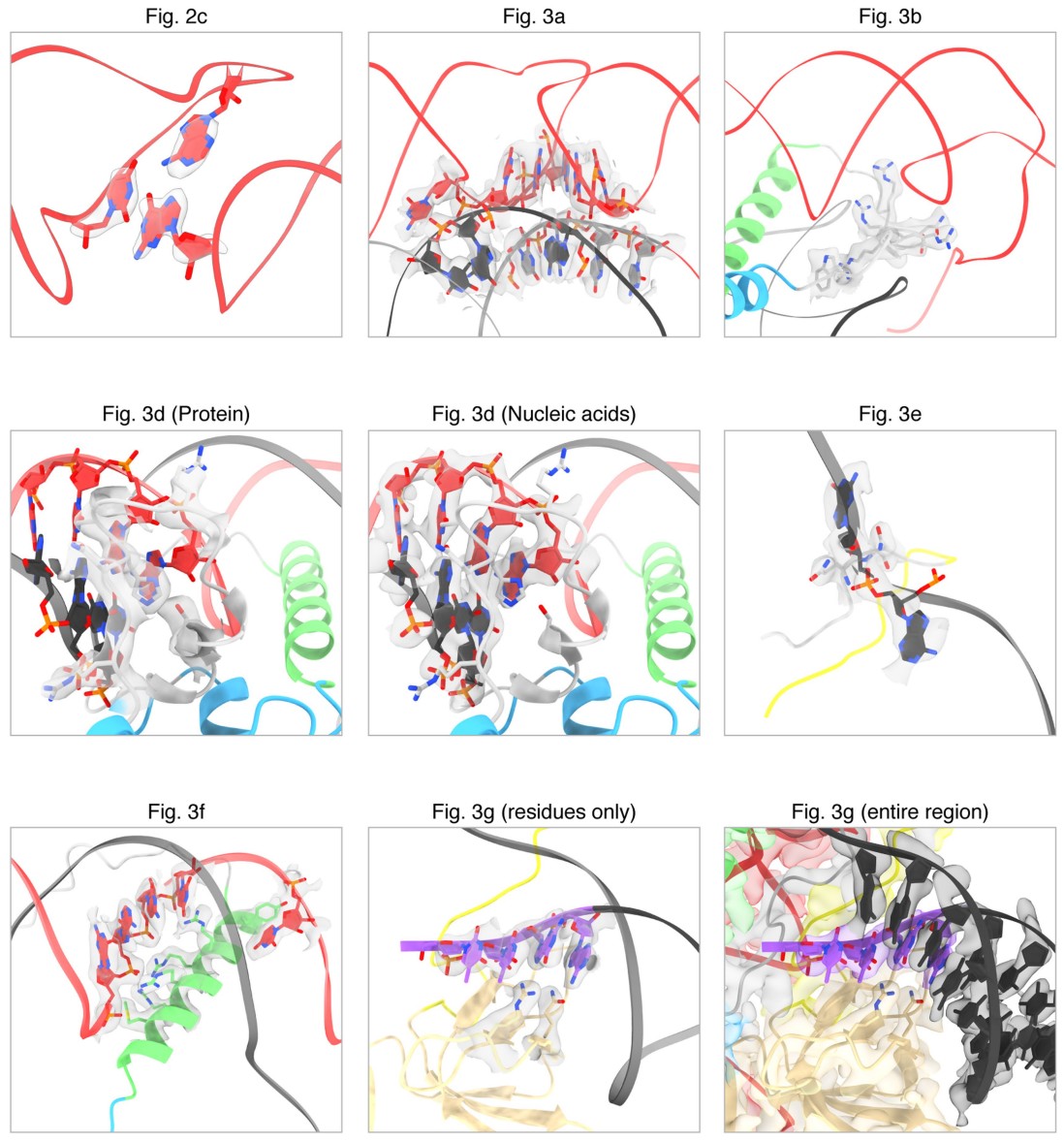

**Extended Data Fig. 2 | Cryo-EM density maps.** Cryo-EM density maps for residues represented in main figures.

**a**

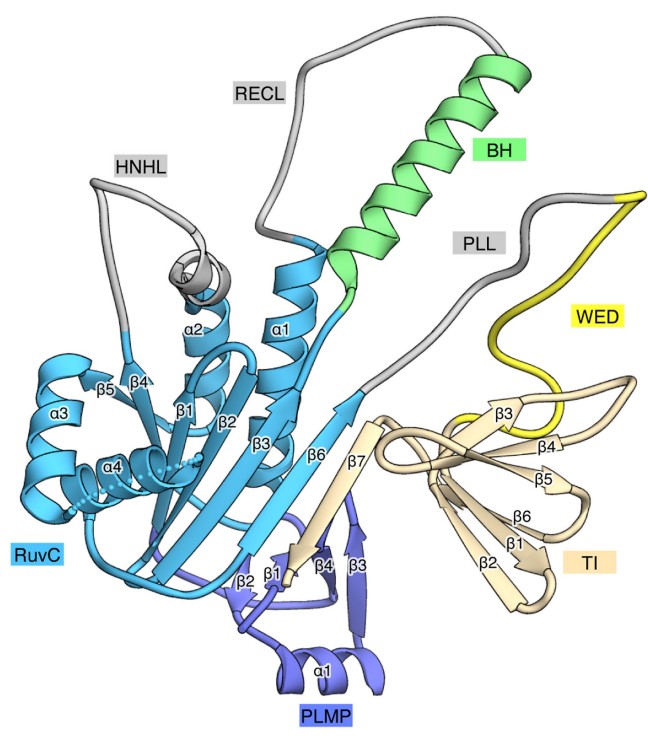

**b**

DtIsrB  SpCas9

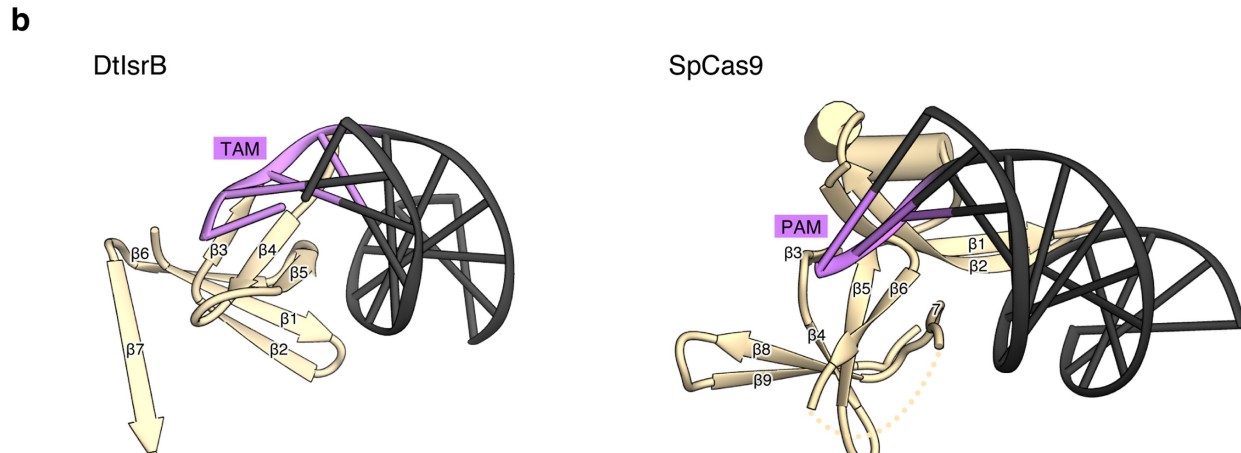

**Extended Data Fig. 3 | Details of the IsrB protein structure.** (a) Close-up view of the IsrB protein structure. (b) Structural comparison between DNA-bound IsrB-TI domain and SpCas9-PI domain (PDB: 7S4X). In the Cas9 structure, the subdomain inserted between β6 and β7 is omitted for clarity.

**a**

| Nr | Hit | Name | Probability | E-value | Score | SS | Aligned cols | Target Length |
|----|-----|------|-------------|---------|-------|-----|--------------|---------------|
| ☐ 1 | PF14239.9 | ; RRXRR ; RRXRR protein | 99.67 | 2.7e-16 | 96.14 | 5.9 | 44 | 173 |
| ☐ 2 | PF20090.2 | ; DUF6482 ; Family of unknown function (DUF6482) | 56.42 | 43 | 18.9 | 2.9 | 26 | 99 |
| ☐ 3 | cd14836 | AP2_Mu_N; AP-2 complex subunit mu N-terminal domain. AP-2 complex mu subunit is part of the heterotetrameric adaptor pro | 52.6 | 24 | 19.4 | 1.5 | 15 | 140 |
| ☐ 4 | 1TIF_A | TRANSLATION INITIATION FACTOR 3; IF3 N-TERMINAL DOMAIN, RIBOSOME BINDING FACTOR; 1.8A {Geobacillus stearothermophilus} S | 52.21 | 70 | 17.52 | 3.5 | 33 | 78 |
| ☐ 5 | PF18225.4 | ; AbfS_sensor ; Sensor histidine kinase (AbfS) sensor domain | 50.71 | 27 | 19.37 | 1.4 | 12 | 65 |

**b**

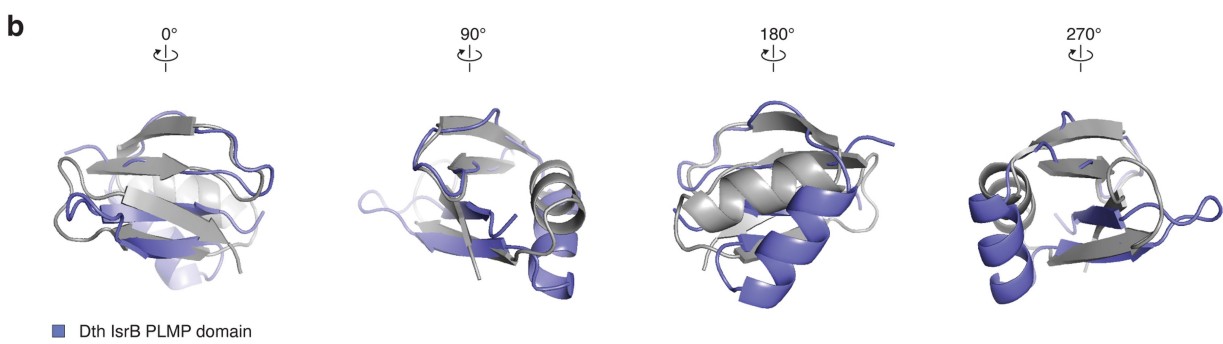

■ Dth IsrB PLMP domain
■ Translation Initiation Factor 3 N-terminal domain (pdb: 1TIF)

**c**

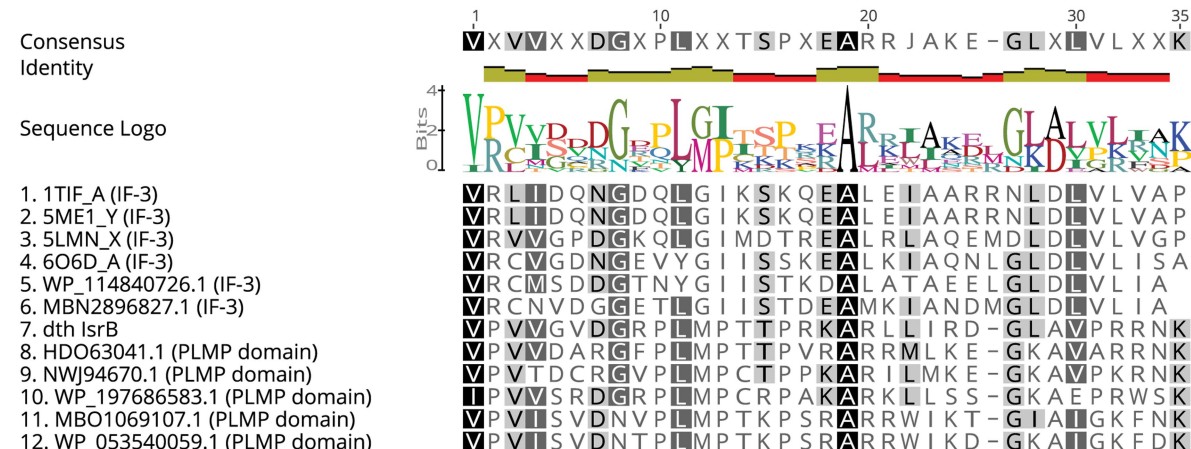

**Extended Data Fig. 4 | PLMP domain homology.** (a) Top five hits from HHPred search using seed sequence SITRVPVVGVDGRPLMPTTPRKARLLIRDGLAVPR RNKLGLFYIQMLRPVGTRTQ corresponding to the PLMP domain from DtIsrB. (b) Structural comparison of the PLMP domain from DtIsrB and the N-terminal domain of Translation Initiation Factor 3 (IF-3) from *Geobacillus stearothermophilus* (PDB: 1TIF). (c) Alignment of representative IF-3 N-terminal domains and OMEGA-related PLMP domains.

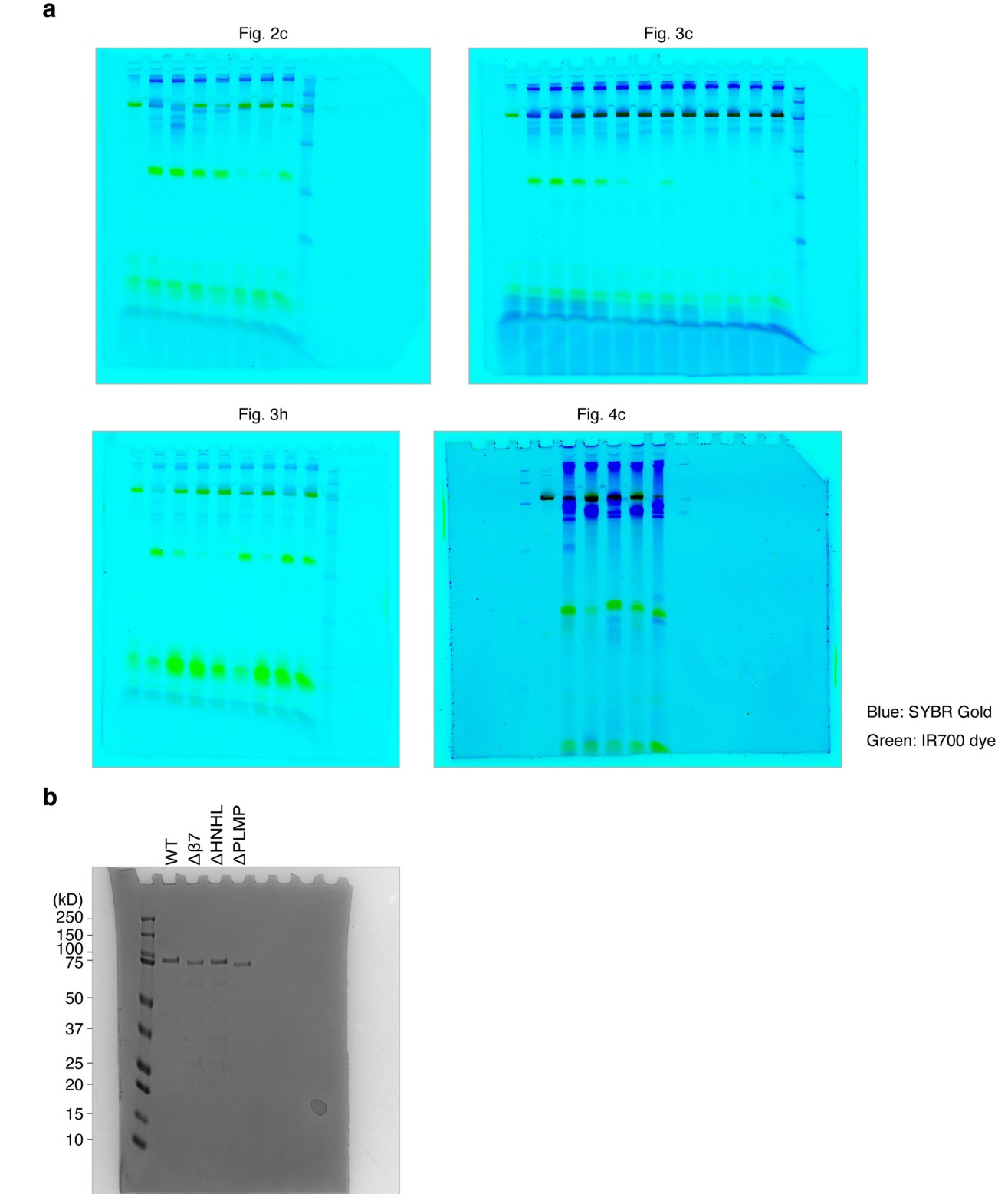

**Extended Data Fig. 5 | Uncropped gel images used in this study.** (a) Denatured PAGE gels for resolving nicked DNA products. (b) An SDS-PAGE gel for expression check of the deletion mutants. Related to Fig. 3c.

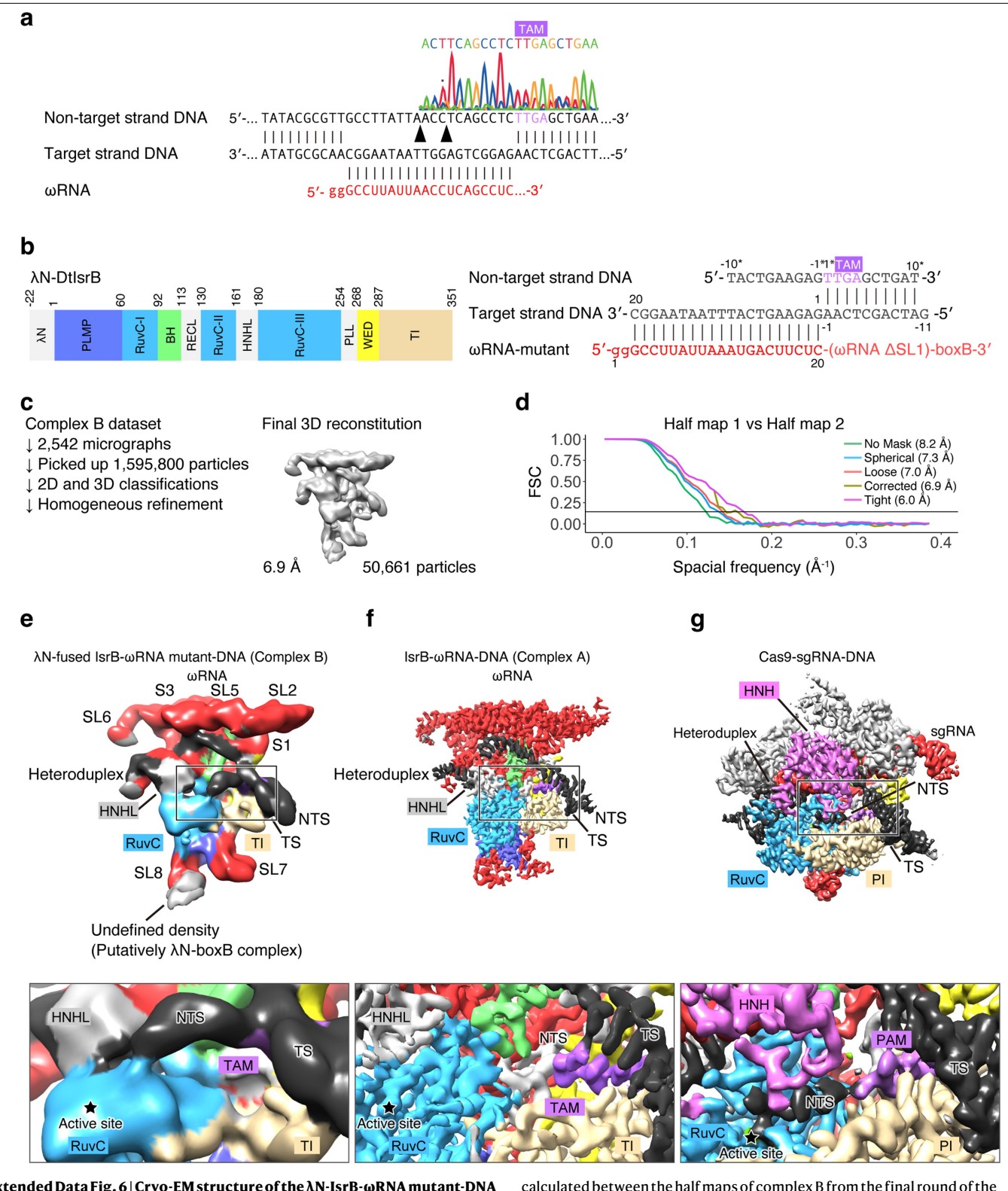

**Extended Data Fig. 6 | Cryo-EM structure of the λN-IsrB-ωRNA mutant-DNA complex (complex B).** (a) Cleavage sites in the target DNA as assay by Sanger sequencing. The nicking sites are marked by black triangles. The additional non-templated adenine is indicated by an asterisk in the Sanger sequencing trace. (b) Domain structure of the λN-IsrB fusion protein (left) and schematic of the ωRNA mutant and target DNA (right). In the ωRNA mutant, residues 34–67 were replaced with GAAA and boxB RNA was appended to the 3' end of the ωRNA scaffold. (c) Cryo-EM data processing schematic for single particle analysis of complex B (Left). Final refined map (Right). (d) FSC curves calculated between the half maps of complex B from the final round of the refinement in cryoSPARC v3.3. (e) Cryo-EM density map of complex B. Based on the superposition of complex B map and complex A model, regions of protein, RNA, and DNA were assigned. Extra density was observed in the vicinity of the ωRNA SL8 region and assigned to the λN-boxB complex, consistent with the SL8-boxB connectivity and the λN-boxB volume (PDB: 1QFQ). TS, target strand; NTS, non-target strand. (f and g) Cryo-EM density maps of complex A (f) and SpCas9 in complex with its cognate RNA and target DNA (EMD: 24838) (g).

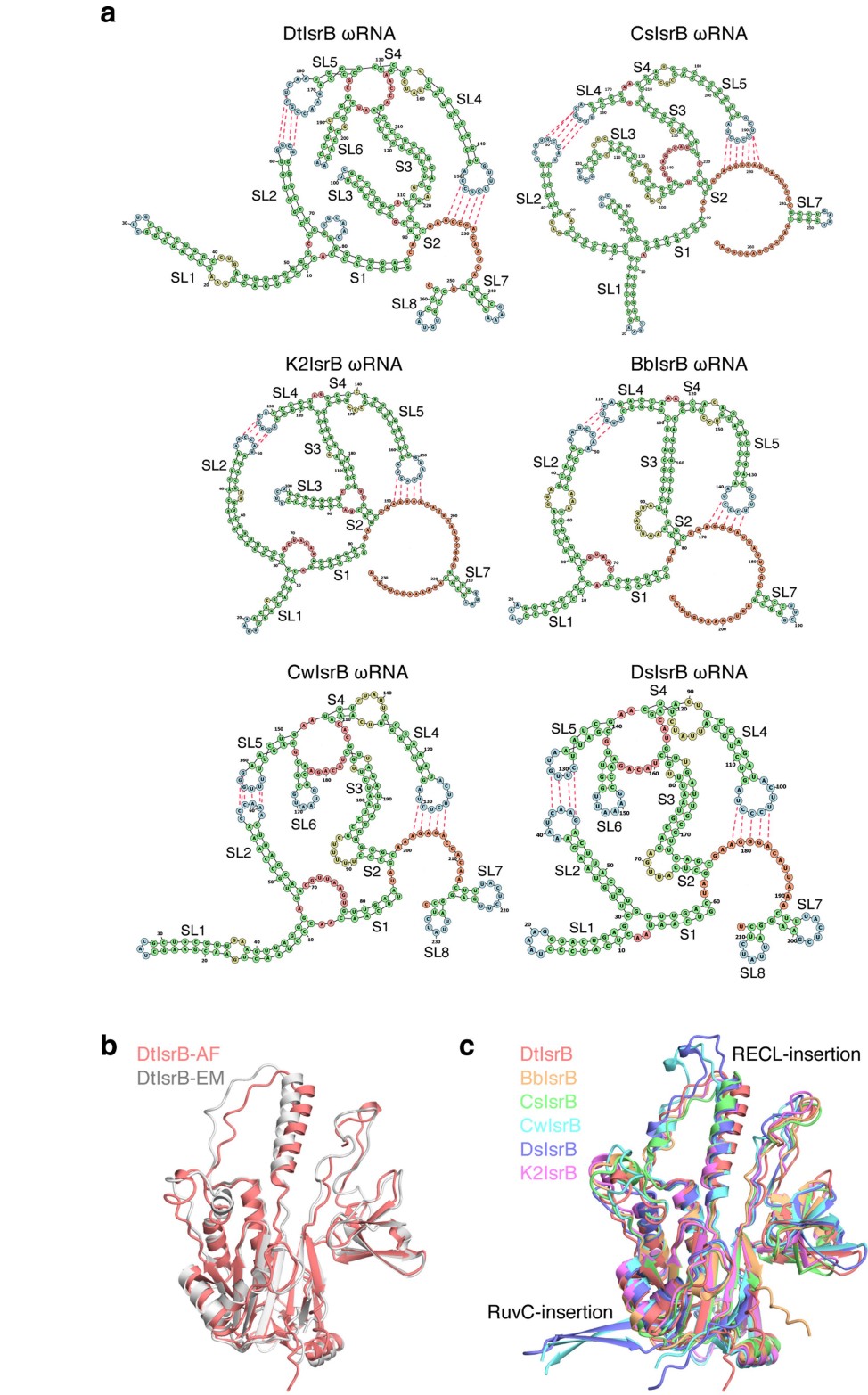

**Extended Data Fig. 7 | Structural prediction of IsrB orthologs and their cognate ωRNAs.** (a) Secondary structure and pseudoknot prediction of the ωRNA scaffolds based on covariance model. In CwIsrB/DsIsrB/BbIsrB ωRNAs, SL3 motifs are replaced with unpaired nucleotides. In CsIsrB/K2IsrB/BbIsrB ωRNAs, SL6 motifs are degenerated and SL8 motifs are replaced with unpaired nucleotides. (b) Superposition of AlphaFold (AF) and cryo-EM (EM) structures of DtIsrB. (c) Superposition of AlphaFold structures of six IsrB orthologs. CwIsrB/DsIsrB have β-hairpin and loop insertions in the RuvC domain and RECL, respectively.

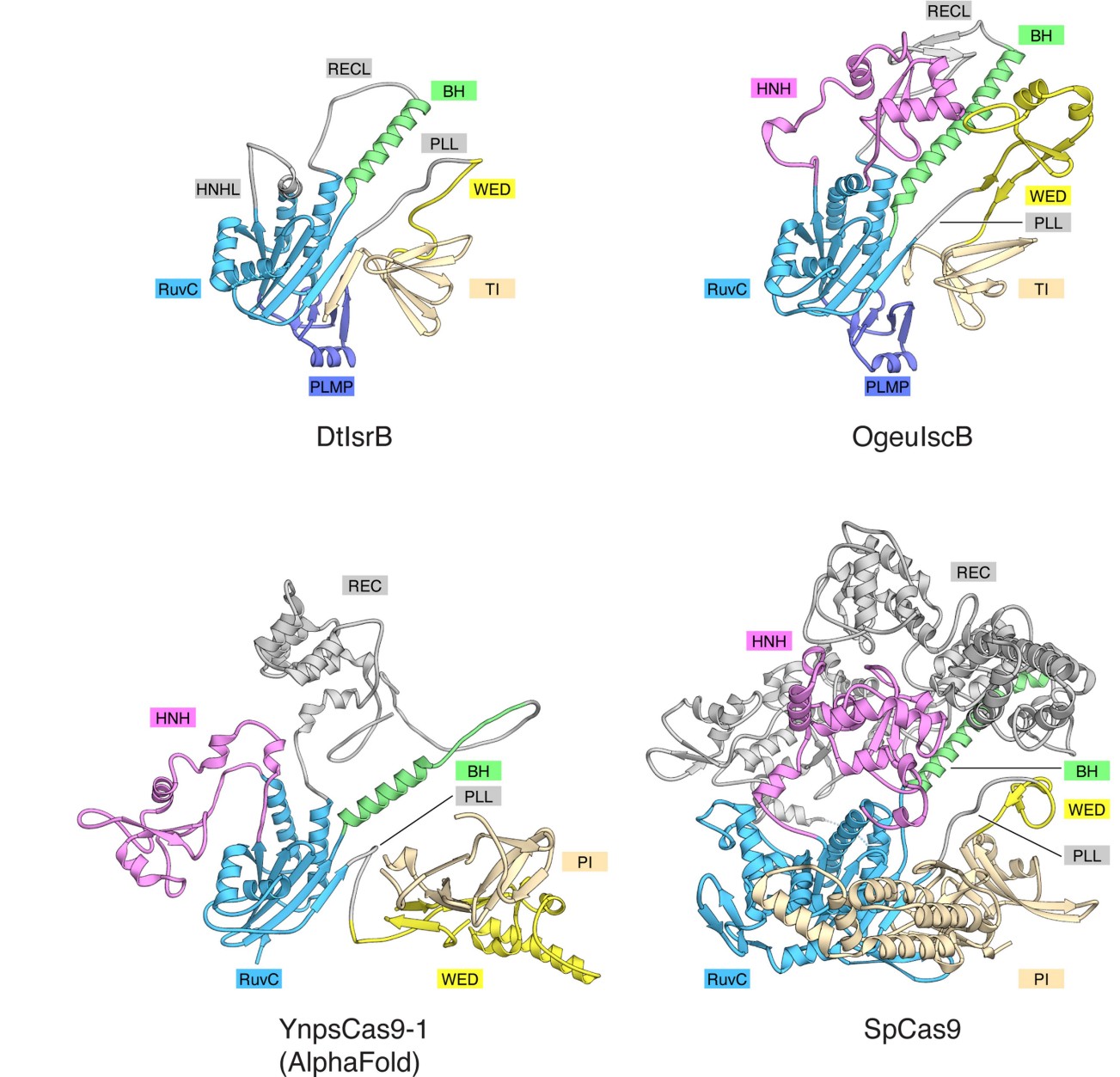

**Extended Data Fig. 8 | Evolutionary snapshot of Cas9 ancestors.** Structural comparison between DtIsrB, OgeuIscB (8CSZ), YnpsCas9-1 (AF2 model), and SpCas9 (PDB: 4OO8).

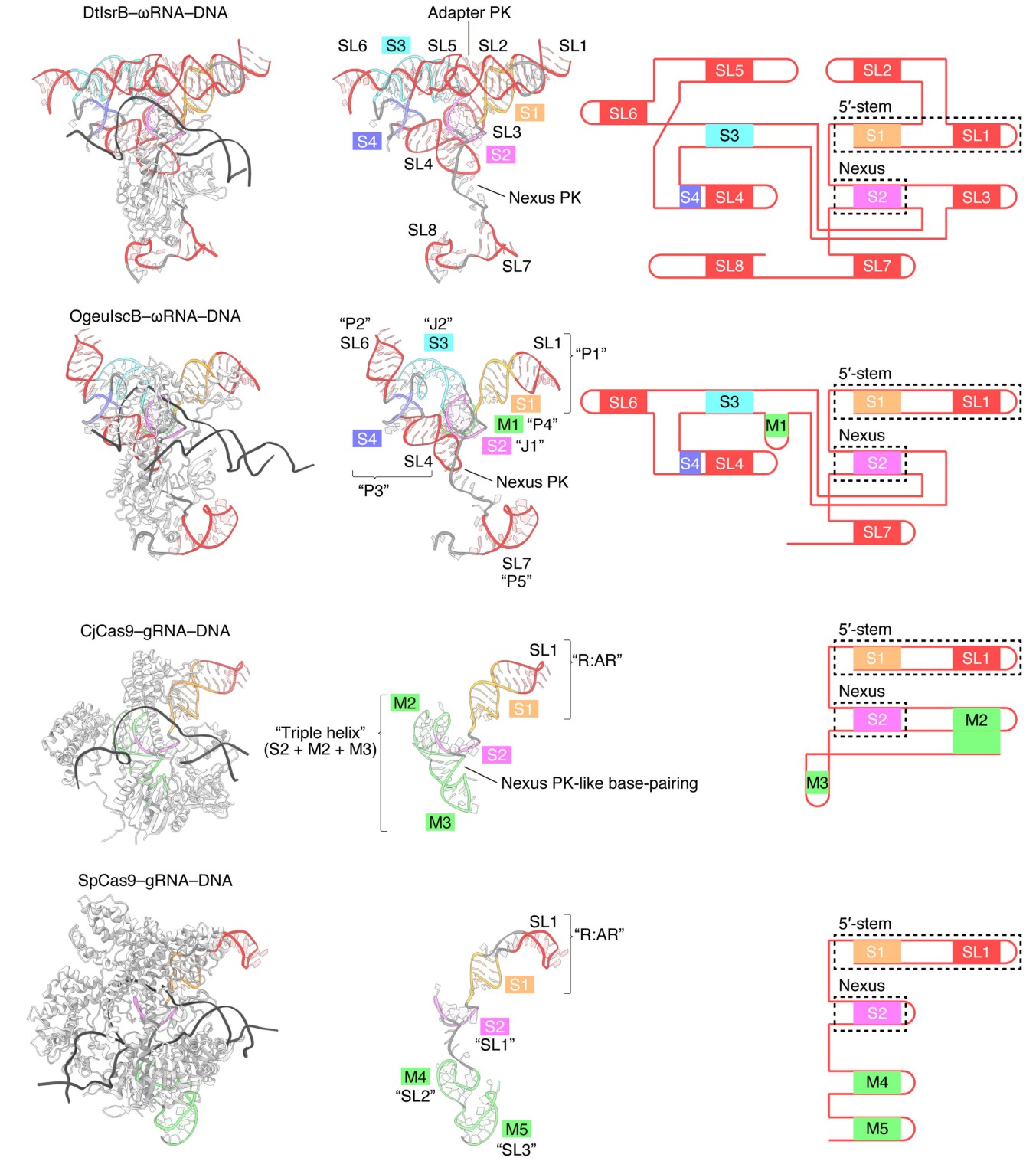

**Extended Data Fig. 9 | Evolutionary snapshot of tracrRNA ancestors.** Structural comparison between cognate RNAs of DtIsrB, OgeuIscB (8CSZ), CjCas9 (5X2G), and SpCas9 (7S4X) in their protein/DNA-bound states. Overall structures (left). RNA structures (center). RNA schematic diagrams (right).

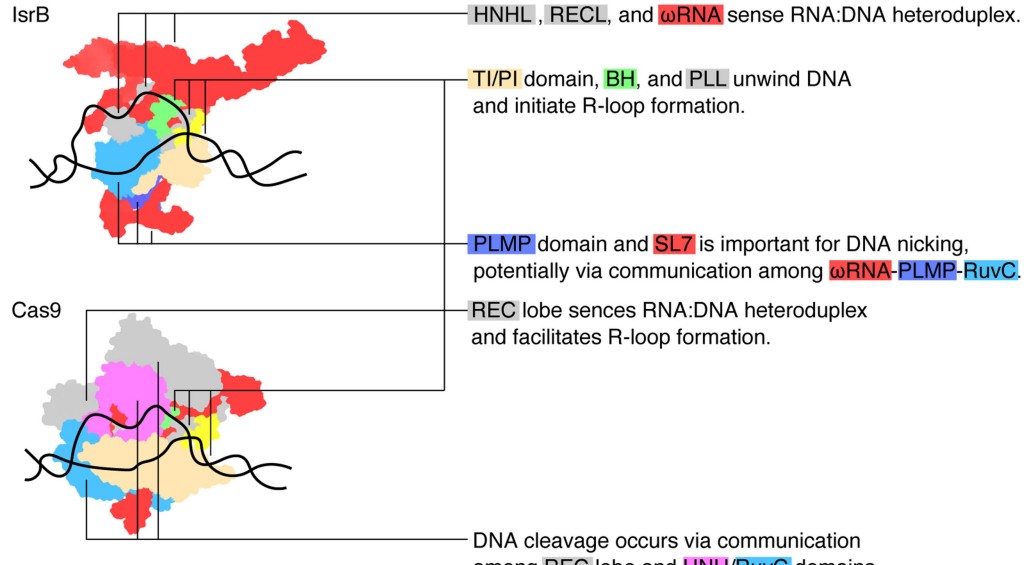

IsrB

HNHL , RECL, and ωRNA sense RNA:DNA heteroduplex.

TI/PI domain, BH, and PLL unwind DNA
and initiate R-loop formation.

PLMP domain and SL7 is important for DNA nicking,
potentially via communication among ωRNA-PLMP-RuvC.

Cas9

REC lobe sences RNA:DNA heteroduplex
and facilitates R-loop formation.

DNA cleavage occurs via communication
among REC lobe and HNH/RuvC domains.

**Extended Data Fig. 10 | Models of RNA-guided DNA nicking/cleavage by IsrB/Cas9.** Schematic highlighting the mechanistic similarities and differences between IsrB and Cas9.

**Extended Data Table 1 | Cryo-EM data collection, refinement, and validation statistics**

|  | Complex A (EMDB-27533) (PDB 8DMB) | Complex B (EMDB-26723) |
|---|---|---|
| **Data collection and processing** | | |
| Magnification | 105,000 | 120,000 |
| Voltage (kV) | 300 | 200 |
| Electron exposure (e–/Å$^2$) | 60 | 62.53 |
| Defocus range (μm) | −2.2 to −0.8 | −2.6 to −1.0 |
| Pixel size (Å) | 0.839 | 1.255 |
| Symmetry imposed | $C1$ | $C1$ |
| Initial particle images (no.) | 1,626,574 | 1,595,000 |
| Final particle images (no.) | 58,188 | 50,661 |
| Map resolution (Å) | 3.1 | 6.9 |
| FSC threshold: 0.143 | | |
| Map resolution range (Å) | 3.1 to 8.9 | 6.0 to 13.8 |
| | | |
| **Refinement** | | N/A |
| Initial model used (PDB code) | N/A | |
| Model resolution (Å) | 3.2 | |
| FSC threshold: 0.5 | | |
| Map sharpening $B$ factor (Å$^2$) | −73.1 | |
| Model composition | | |
| Non-hydrogen atoms | 8497 | |
| Protein residues | 328 | |
| Nucleotide residues | 275 | |
| Ligands | 2 | |
| $B$ factors (Å$^2$) | | |
| Protein | 24.60 | |
| Nucleotide | 62.18 | |
| Ligand | 21.56 | |
| R.m.s. deviations | | |
| Bond lengths (Å) | 0.005 | |
| Bond angles (°) | 0.902 | |
| Validation | | |
| MolProbity score | 0.86 | |
| Clashscore | 0.87 | |
| Poor rotamers (%) | 0.74 | |
| Ramachandran plot | | |
| Favored (%) | 97.53 | |
| Allowed (%) | 2.47 | |
| Disallowed (%) | 0.00 | |

# Reporting Summary

## Statistics

For all statistical analyses, confirm that the following items are present in the figure legend, table legend, main text, or Methods section.

| n/a | Confirmed | |
|---|---|---|
| ☒ | ☐ | The exact sample size (*n*) for each experimental group/condition, given as a discrete number and unit of measurement |
| ☒ | ☐ | A statement on whether measurements were taken from distinct samples or whether the same sample was measured repeatedly |
| ☒ | ☐ | The statistical test(s) used AND whether they are one- or two-sided *Only common tests should be described solely by name; describe more complex techniques in the Methods section.* |
| ☒ | ☐ | A description of all covariates tested |
| ☒ | ☐ | A description of any assumptions or corrections, such as tests of normality and adjustment for multiple comparisons |
| ☒ | ☐ | A full description of the statistical parameters including central tendency (e.g. means) or other basic estimates (e.g. regression coefficient) AND variation (e.g. standard deviation) or associated estimates of uncertainty (e.g. confidence intervals) |
| ☒ | ☐ | For null hypothesis testing, the test statistic (e.g. *F*, *t*, *r*) with confidence intervals, effect sizes, degrees of freedom and *P* value noted *Give P values as exact values whenever suitable.* |
| ☒ | ☐ | For Bayesian analysis, information on the choice of priors and Markov chain Monte Carlo settings |
| ☒ | ☐ | For hierarchical and complex designs, identification of the appropriate level for tests and full reporting of outcomes |
| ☒ | ☐ | Estimates of effect sizes (e.g. Cohen's *d*, Pearson's *r*), indicating how they were calculated |

*Our web collection on statistics for biologists contains articles on many of the points above.*

## Software and code

Policy information about availability of computer code

| Data collection | EPU v2.12.1, SerialEM v4.0.4 |
|---|---|
| Data analysis | MotionCor2, CTFFIND4.1, RELION4.0, cryoSPARC v3.3, ColabFold v1.3.0, Auto-DRRAFTER distributed in Rosetta 3.13, MapQ v1.8.2, ISOLDE v1.0b3, PHENIX v1.20.1, COOT v0.9.6, UCSF Chimera v1.15, ChimeraX v1.4, Cuemol2 v2.2.3.443, PyMOL v2.5.2, cmsearch distributed in brendanf/inferrnal v0.99.6, mfold (http://www.unafold.org/), Forna distributed in ViennaRNA Package 2.5, custom python script for TAM analysis (https://zenodo.org/record/5168777#.YmM5iPOZOJM) |

For manuscripts utilizing custom algorithms or software that are central to the research but not yet described in published literature, software must be made available to editors and reviewers. We strongly encourage code deposition in a community repository (e.g. GitHub). See the Nature Portfolio guidelines for submitting code & software for further information.

## Data

Policy information about availability of data

All manuscripts must include a data availability statement. This statement should provide the following information, where applicable:

- Accession codes, unique identifiers, or web links for publicly available datasets
- A description of any restrictions on data availability
- For clinical datasets or third party data, please ensure that the statement adheres to our policy

The coordinate of IsrB ternary structure: PDB 8DMB. The EM map of complex A: EMD27533. The EM map of complex B: EMD26723.

# Human research participants

Policy information about studies involving human research participants and Sex and Gender in Research.

| | |
|---|---|
| Reporting on sex and gender | NA. Human research participants are not involved in this study. |
| Population characteristics | NA. Human research participants are not involved in this study. |
| Recruitment | NA. Human research participants are not involved in this study. |
| Ethics oversight | NA. Human research participants are not involved in this study. |

Note that full information on the approval of the study protocol must also be provided in the manuscript.

# Field-specific reporting

Please select the one below that is the best fit for your research. If you are not sure, read the appropriate sections before making your selection.

☒ Life sciences          ☐ Behavioural & social sciences          ☐ Ecological, evolutionary & environmental sciences

For a reference copy of the document with all sections, see nature.com/documents/nr-reporting-summary-flat.pdf

# Life sciences study design

All studies must disclose on these points even when the disclosure is negative.

| | |
|---|---|
| Sample size | For complex A/B data analyses, 4142/2542 micrographs and 58188/50661 final particle images are used, respectively. |
| Data exclusions | To obtain the higher resolution image of cryo-EM reconstitution, the damaged and 'bad' particles are excluded, based on the software algorithm of single-particle analysis. |
| Replication | Cryo-EM data processing methods used in RELION and cryoSPARC softwares are well-established and reproducible. DNA nicking experiments are repeated at least three times and confirmed for their reproducibility. |
| Randomization | NA |
| Blinding | NA |

# Reporting for specific materials, systems and methods

We require information from authors about some types of materials, experimental systems and methods used in many studies. Here, indicate whether each material, system or method listed is relevant to your study. If you are not sure if a list item applies to your research, read the appropriate section before selecting a response.

## Materials & experimental systems

| n/a | Involved in the study |
|---|---|
| ☒ | ☐ Antibodies |
| ☒ | ☐ Eukaryotic cell lines |
| ☒ | ☐ Palaeontology and archaeology |
| ☒ | ☐ Animals and other organisms |
| ☒ | ☐ Clinical data |
| ☒ | ☐ Dual use research of concern |

## Methods

| n/a | Involved in the study |
|---|---|
| ☒ | ☐ ChIP-seq |
| ☒ | ☐ Flow cytometry |
| ☒ | ☐ MRI-based neuroimaging |

