## [Peer Review File · Nature]

Manuscript Title: Structure of the OMEGA nickase IsrB in complex with ω RNA and target DNA

Reviewer Comments & Author Rebuttals

Reviewer Reports on the Initial Version:

Referees' comments:

Referee #1:

This manuscript reports the cryo-EM structure of the transposon-encoded nickase IsrB in complex with an omega guide RNA and target DNA. Although the function of IsrB is not well understood, IsrB and the related protein, IscB, are thought to be ancestral to the Cas9 enzyme, so there is particular interest in them, particularly given their small size. The structure is solved at an overall resolution of 4.3 Å, but there is considerable variation in resolution of different parts; in particular, much of the RNA is observed at a much lower resolution. The low resolution complicates structural assignment of the omega RNA and this should be checked carefully by a cryoEM specialist. Structural features of the omega RNA are probed by mutagenesis to identify essential and non-essential elements, although these experiments do not give a major new insight into the mechanism of the nickase or target DNA recognition.

The structure reveals interesting similarities between IsrB and Cas9, with the noted absence of the entire REC domain in the former. Extensive SDM has been carried out to test the roles of various aa residues implicated in target DNA recognition and cleavage. Perhaps most interesting is the dissection of the molecular basis for TAM (target adjacent motif?) recognition. Such details are of course highly specific to individual systems, but could aid in engineering new TAMs. The final section focuses on the trajectory of evolution from RuvC to Cas9, which may be a highlight of the paper for many readers.

Overall, this is quite a densely written manuscript with a lot of structural data. The findings are certainly interesting, and the work is strongly complimentary to the accompanying paper.

Specific points:

1. Line 38: Need to define "TAM".
2. Line 153: Deletion of the HNHL (aa 161-174) is reported to abolish nicking activity. This is a major edit of the protein. Is the variant still folded/stable, and if so how was this assessed? There is a risk that this modification destabilises the protein, resulting in a trival reason for non-functionality.
3. I could find no mention of any statistical treatment for the biochemical data in Figs 2-4. How often were these repeated? There is also a lack of any marker lane on the gels in these figures and the sizes of substrate and product are not indicated in the figures.

4. It might be worth stating that *Desulfovibrio thermocuniculi* is a thermophilic organism, explaining the need for biochemical assays at 60 °C.

5. Although the evolution of the protein component is discussed in detail, there is no equivalent consideration of the evolution of omega RNA to tracrRNA. This would add to the general interest of the paper.

Referee #2:

Hirano et al. report the cryo-EM structure of the nickase, IsrB, in complex with ω RNA and target DNA. IsrB is a homolog of IscB, the likely ancestor of Cas9, and lacks the HNH nuclease domain. IsrB (~350 amino acids long) is very small compared to Cas9 but counterbalanced by a relatively large guide RNA (~300-nt ω RNA) for RNA-guided target DNA recognition. The authors determined the cryo-EM structure of an IsrB- ω RNA-target DNA complex. They found that IsrB relies on ω RNA to facilitate target selection.

Overall, this is a nice paper and enjoyable to read. However, the major weakness is the limited resolution of the cryo-EM map. The functional studies were largely based on a structural model, which was built by a combination of approaches including an intermediate resolution cryo-EM map and computational methods such as AlphaFold2 and auto-DRRAFTER. Although mutagenesis results are in general consistent with the claimed interactions, the limited resolution of the cryo-EM map raises concerns about the conclusions in this paper. It is not clear from the writing in this paper, what prevents a higher resolution structure of the complex.

Specific comments:

1) Fittings between individual amino acid residues and the cryo-EM map are not shown. The authors showed an FSC curve comparing the final map and the final model, indicating a resolution at 4.7 Å (Extended Data Figure 1e). At this resolution, many amino acid side chains are not visible.

2) The PDB validation reports were preliminary ones and not for manuscript review.

3) TAM was not spelled out at the first time.

4) It is interesting to observe that β 7 extensively interacts with the core fold of the TI domain. Is β 7 functionally important?

5) The authors stated that 'The convergence of these models is 4.0 Å, and based on this value, the estimated accuracy of the models is 4.9 Å'. How was the accuracy determined? What does it exactly mean?

6) Can the authors show the original maps that support interactions shown in Fig. 3 f-h?

7) This sentence is difficult to follow, possibly because of too condensed information. Consider

revising it. 'This three-dimensional feature is realized by the ω RNA primary sequence (S1-SL1-SL2-S1-S2-SL3-S3-S4-SL4-S4-SL5-SL6-S3-S2-SL7-SL8 [Unpaired nucleotides are omitted. The eight basepairing regions are underlined.]) (Fig. 2a).'

8) The author described communication among three layers in ω RNA. What is the exact meaning of communications here? It looks there are just contacts.

9) What is the guide adapter region? An explanation is required.

10) Add the guide region in Fig. 2a.

11) In Fig. 4c, the amounts of uncut substrate are clearly different. Was this a result of loading errors?

12) Many discussions are based on the assumption that IscB is the ancestor of Cas9. Can the authors comment on the confidence of IscB being the ancestor of Cas9?

13) An extremely low concentration (0.05 mg/ml) was used for cryo-EM grid preparation. Was there a support film on the grids?

14) PDB and EMD IDs were not included.

Referee #3:

Hirano et al. determine the structure of IscB in complex with the guide RNA and target DNA. This is an especially exciting story and a great follow up to the RNP. This study provides HINTS at really interesting insights. It's unfortunate that the structure is at moderate resolution, which requires many of the reported insights to be supported by more biochemical data. The quality of the maps simply does not support unambiguous modeling. This reviewer cannot recommend publication in Nature, as it stands. The manuscript needs a significant number of additional experiments to be performed before consideration.

Major comments:

1. The actual roles of residues mentioned in the manuscript must be demonstrated through mutagenesis and biochemical assays. This has been done for some, but needs to be expanded upon for any residues for which the authors based their interpretation. The structure itself is not unambiguous enough to base claims.

2. Greater than 12 million particles were picked in A2 dataset – but only ~120k ended up in the final reconstruction. Was the prep bad? Please include representative micrographs and 2D class averages. While it is common to discard ~90% of particles for a final reconstruction, this is somewhat concerning. 20x more micrographs in A2 vs A1, but only 2x as many particles in final reconstruction?

3. In the results section, lines 48-49 “we manually built...”: The authors could not have possibly manually built this. I suspect they fitted the AlphaFold (AF2) model into their low-resolution map.

4. The TAM preference assay – comparison between TTGA and ATGG: It is not possible to deduce what difference positions 1 or 4 make for Q326R. It would be more appropriate to test TTGG instead of ATGG, since the current data does not allow one to draw direct comparisons since two residues are changed simultaneously.

5. Please show representative densities for all regions shown in Fig. 3e. Since the resolution of the map is low, it is necessary to test the role of BH Arg residues in IsrB function.

6. The interactions between DNA and RNA backbones are not convincing. Such features are common for low-resolution cryo-EM maps – if one were to go to a sufficient threshold, interactions start to appear. This reviewer would suggest removal of this section.

7. The entire region of density should be shown in Ext. Data Fig. 1g, rather than volume around only R323 and Q326.

8. Please show density for HNHL.

9. Where is the comparison between Δ SL1 and full-length complex? There is no direct comparison.

Minor comment:

1. It is not surprising that the RNA makes little contact with targeted DNA, aside from the spacer base-pairing with the target DNA. This is a puzzling parallel to make since it seems rather obvious based on our knowledge of RNA guide - DNA interactions. It's possible that this reviewer is missing something?

David Taylor

Author Rebuttals to Initial Comments:

Referee #1:

This manuscript reports the cryo-EM structure of the transposon-encoded nickase IsrB in complex with an omega guide RNA and target DNA. Although the function of IsrB is not well understood, IsrB and the related protein, IscB, are thought to be ancestral to the Cas9 enzyme, so there is particular interest in them, particularly given their small size. The structure is solved at an overall resolution of 4.3 Å, but there is considerable variation in resolution of different parts; in particular, much of the RNA is observed at a much lower resolution. The low resolution complicates structural assignment of the omega RNA and this should be checked carefully by a cryoEM specialist. Structural features of the omega RNA are probed by mutagenesis to identify essential and non-essential elements, although these experiments do not give a major new insight into the mechanism of the nickase or target DNA recognition.

The structure reveals interesting similarities between IsrB and Cas9, with the noted absence of the entire REC domain in the former. Extensive SDM has been carried out to test the roles of various aa residues implicated in target DNA recognition and cleavage. Perhaps most interesting is the dissection of the molecular basis for TAM (target adjacent motif?) recognition. Such details are of course highly specific to individual systems, but could aid in engineering new TAMs. The final section focuses on the trajectory of evolution from RuvC to Cas9, which may be a highlight of the paper for many readers.

Overall, this is quite a densely written manuscript with a lot of structural data. The findings are certainly interesting, and the work is strongly complimentary to the accompanying paper.

We thank the Reviewer for the positive feedback!

Specific points:

1. Line 38: Need to define "TAM".

We have revised this.

2. Line 153: Deletion of the HNHL (aa 161-174) is reported to abolish nicking activity. This is a major edit of the protein. Is the variant still folded/stable, and if so how was this assessed? There is a risk that this modification destabilises the protein, resulting in a trival reason for non-functionality.

To confirm the stability of proteins produced by the deletion mutants, we demonstrated

that all these proteins were present in the bacterial lysate overexpressing the deletion mutants (Extended Data Fig. 5b).

3. I could find no mention of any statistical treatment for the biochemical data in Figs 2-4. How often were these repeated? There is also a lack of any marker lane on the gels in these figures and the sizes of substrate and product are not indicated in the figures.

We have repeated the biochemical assay three times. We have shown complete, representative gels in Extended Data Fig. 5a. The product and substrate sizes are measured by the superposition with SYBR-Gold stained RNA ladder marker.

4. It might be worth stating that *Desulfovibrio thermocuniculi* is a thermophilic organism, explaining the need for biochemical assays at 60 °C.

We apologize for this oversight. This has been added in the main text (in the description of the TAM preference) as well as in the methods section.

5. Although the evolution of the protein component is discussed in detail, there is no equivalent consideration of the evolution of omega RNA to tracrRNA. This would add to the general interest of the paper.

We thank the Reviewer for this helpful suggestion. We have added a discussion on the evolution of the ω RNA to cr/tracrRNA (Extended Data Fig. 9).

Referee #2:

Hirano et al. report the cryo-EM structure of the nickase, IsrB, in complex with ω RNA and target DNA. IsrB is a homolog of IscB, the likely ancestor of Cas9, and lacks the HNH nuclease domain. IsrB (~350 amino acids long) is very small compared to Cas9 but counterbalanced by a relatively large guide RNA (~300-nt ω RNA) for RNA-guided target DNA recognition. The authors determined the cryo-EM structure of an IsrB- ω RNA-target DNA complex. They found that IsrB relies on ω RNA to facilitate target selection.

Overall, this is a nice paper and enjoyable to read. However, the major weakness is the limited resolution of the cryo-EM map. The functional studies were largely based on a structural model, which was built by a combination of approaches including an intermediate resolution cryo-EM map and computational methods such as AlphaFold2 and auto-DRRAFTER. Although mutagenesis results are in general consistent with the claimed interactions, the limited resolution of the cryo-EM map raises concerns about the conclusions in this paper. It is not clear from the writing in this paper, what prevents a higher resolution structure of the complex.

We thank the Reviewer for the positive comments. To address the major weakness pointed

out, we optimized our approach to improve the resolution of the cryo-EM map. Specifically, by adding magnesium and a crosslinking reagent, we overcame the orientation bias and ω RNA flexibility issues. The overall resolution was improved from 4.3 Å to 3.1 Å. The improved resolution reinforced our conclusions, and we believe it addresses the concerns raised about model ambiguity.

Specific comments:

1) Fittings between individual amino acid residues and the cryo-EM map are not shown. The authors showed an FSC curve comparing the final map and the final model, indicating a resolution at 4.7 Å (Extended Data Figure 1e). At this resolution, many amino acid side chains are not visible.

In the current map, with the improved resolution, many amino acid side chains are visible (Extended Data Fig. 2).

2) The PDB validation reports were preliminary ones and not for manuscript review.

We apologize that the official PDB validation reports were unavailable, due to the tight scheduling of the submission of our manuscript. For the revised structure, we contacted the PDB on July 8th about re-submission of the coordinate and map files. As soon as we get the revised validation reports, we will send them to the editorial office for forwarding for review.

3) TAM was not spell out at the first time.

We have revised this.

4) It is interesting to observe that β 7 extensively interacts with the core fold of the TI domain. Is β 7 functionally important?

We thank the Reviewer for raising this question. We have now performed additional biochemical analyses and found that truncation of β 7 abolished DNA nicking activity of IsrB (Fig. 3c), indicating that it is functionally important.

5) The authors stated that 'The convergence of these models is 4.0 Å, and based on this value, the estimated accuracy of the models is 4.9 Å'. How was the accuracy determined? What does it exactly mean?

We apologize for the lack of clarity about this point. In the revised manuscript, we have clarified the definition of convergence (pairwise root mean square deviation between models) and the relationship between convergence and auto-DRRAFTER model accuracy in

the “Model building and validation” section of the Methods. Previously, it was shown that there is a linear relationship between auto-DRRAFTER modeling convergence and model accuracy (i.e., root mean square deviation to the “true” coordinates), and that this relationship can be used to estimate auto-DRRAFTER model accuracy (Kappel, et al. Nature Methods, 2020). However, we note that this estimate reflects only the accuracy of the initial auto-DRRAFTER models. With our new 3.1 Å resolution density map, it is possible to manually refine the RNA coordinates, thus improving the accuracy of our final model.

6) Can the authors show the original maps that support interactions shown in Fig. 3 f-h?
We have added the suggested EM-density maps to Extended Data Fig 2.

7) This sentence is difficult to follow, possibly because of too condensed information. Consider revising it. ‘This three-dimensional feature is realized by the ω RNA primary sequence (S1-SL1-SL2-S1-S2-SL3-S3-S4-SL4-S4-SL5-SL6-S3-S2-SL7-SL8 [Unpaired nucleotides are omitted. The eight basepairing regions are underlined.]) (Fig. 2a).’

We have revised this sentence for readability.

8) The author described communication among three layers in ω RNA. What is the exact meaning of communications here? It looks there are just contacts.

We thank the Reviewer for pointing this out. We have replaced “communication” with “interaction” in the main text and tried to clarify our thinking. In Cas9, due to the inter-domain communication among REC/HNH/RuvC domains, target sensing by the REC domain is coupled with target cleavage by the RuvC domain. Analogously, in IsrB, due to interactions among layers 1/2/3, target sensing by layer 1 is potentially coupled with target nicking by the RuvC domain, because layer 1 (SL2/4/5) recognizes the guide:target heteroduplex and layer 3 (SL7/8) binds to the PLMP/RuvC/TI domains.

9) What is the guide adapter region? An explanation is required.

We apologize for this oversight. We have now explained this in the maintext, as follows:

“The 5'-stem region of ω RNA (S1, SL1, and SL2) is designated the guide adaptor. It appears that during the evolutionary transition from the OMEGA system to CRISPR-Cas, SL2 and the descending strands of S1/SL1 of the ω RNA were adapted to form the CRISPR array to enable the formation of the functional Cas9-crRNA-tracrRNA complex (Fig. 1a). The genomic sequence encoding the guide adapter region is important for IS200/IS605 transposon activity in bacterial genomes (Fig. 2a).”

10) Add the guide region in Fig. 2a.

We have revised this.

11) In Fig. 4c, the amounts of uncut substrate are clearly different. Was this a result of loading errors?

We agree with the Reviewer that there are differences between the orthologs, although this might reflect intrinsic differences in their enzymatic activity. Nevertheless, we have repeated the experiment and obtained similar results, indicating there was not a loading error. Based on our previous biochemical assay (Altae-Tran et al., *Science*, 2021), for each ortholog, we used 5 nM IsrB template for in vitro transcription/translation reaction (IVTT), 50 nM ω RNA template for IVTT, and 10 nM substrate in the reaction. To boost the signals of weak DNA nicking activities of IsrB orthologs, we incubated the reaction for 2–6 hr (described as in “In vitro cleavage experiment” method section). We found that the K2IsrB substrate was degraded during the 2 hr incubation, likely because K2IsrB seems to have a non-specific DNA degradation activity, again indicating that there are intrinsic differences between various IsrB orthologs.

12) Many discussions are based on the assumption that IscB is the ancestor of Cas9. Can the authors comment on the confidence of IscB being the ancestor of Cas9?

We apologize for the confusion. In our previous work (Altae-Tran et al., *Science*, 2021), we extended and refined a previous analysis (Kapitonov et al., *J. Bacteriology*, 2015) demonstrating that IscB is the ancestor of Cas9 (and that IsrB is the likely antecedent of IscB). This is based both on phylogenetic analysis and on the unique domain architecture shared by IscB and Cas9. Thus, we are confident in this assumption, and our structural work is consistent with this relationship. We have highlighted this more prominently in the introduction.

13) An extremely low concentration (0.05 mg/ml) was used for cryo-EM grid preparation. Was there a support film on the grids?

We thank the Reviewer for this inquiry. In this case, a higher concentration (>0.2 mg/ml) made the grid holes empty/dry. Although we appreciate the suggestion to try the support film, our collaborators suggested another path forward that produced high quality data. We will certainly keep this tip in mind for the future, though!

14) PDB and EMD IDs were not included.

We have updated these.

Referee #3:

Hirano et al. determine the structure of IsrB in complex with the guide RNA and target DNA.

This is an especially exciting story and a great follow up to the RNP. This study provides HINTS at really interesting insights. It's unfortunate that the structure is at moderate resolution, which requires many of the reported insights to be supported by more biochemical data. The quality of the maps simply does not support unambiguous modeling. This reviewer cannot recommend publication in Nature, as it stands. The manuscript needs a significant number of additional experiments to be performed before consideration.

As noted above, we have optimized our approach to improve the resolution of the cryo-EM map. Specifically, by adding magnesium and a crosslinking reagent, we overcame the orientation bias and ω RNA flexibility issues. The overall resolution was improved from 4.3 Å to 3.1 Å. The improved resolution reinforced our conclusions, and we believe supports the functional and evolutionary insights described in the manuscript.

Major comments:

1. The actual roles of residues mentioned in the manuscript must be demonstrated through mutagenesis and biochemical assays. This has been done for some, but needs to be expanded upon for any residues for which the authors based their interpretation. The structure itself is not unambiguous enough to base claims.

We have tested additional mutants (the adapter pseudoknot double mutant, the R100A mutant, the R104A mutant, the F119A/R124A mutant, the β 7 truncation mutant, and the PLMP truncation mutant), to further validate our claims on IsrB-specific mechanisms. Moreover, the improved resolution of the structure supports our claims.

2. Greater than 12 million particles were picked in A2 dataset – but only ~120k ended up in the final reconstruction. Was the prep bad? Please include representative micrographs and 2D class averages. While it is common to discard ~90% of particles for a final reconstruction, this is somewhat concerning. 20x more micrographs in A2 vs A1, but only 2x as many particles in final reconstruction?

We thank the Reviewer for raising this critical point. In the previous dataset, we had to discard many particles after re-extracting particles for high-resolution 3D refinement, potentially due to the high degree of flexibility in the ω RNA. In the current dataset, we optimized the conditions (adding magnesium) to fix the ω RNA, and it worked well (Particle numbers are 1.6M [Autopick], 0.1M [After C3D], and 0.06M [After no-align C3D], values which are in line with the Reviewer's comment).

3. In the results section, lines 48-49 “we manually built...”: The authors could not have

possibly manually built this. I suspect they fitted the AlphaFold (AF2) model into their low-resolution map.

We apologize for the confusion. We have revised this part to reflect that we manually modified the initial protein model generated by AlphaFold for protein structural refinement. We describe this process in the Methods section.

4. The TAM preference assay – comparison between TTGA and ATGG: It is not possible to deduce what difference positions 1 or 4 make for Q326R. It would be more appropriate to test TTGG instead of ATGG, since the current data does not allow one to draw direct comparisons since two residues are changed simultaneously.

We thank the Reviewer for this suggestion. We have now checked the TTGG substrate specificities of wild-type and the Q326 mutant (Fig. 3h).

5. Please show representative densities for all regions shown in Fig. 3e. Since the resolution of the map is low, it is necessary to test the role of BH Arg residues in IsrB function.

We have added the HNHL, heteroduplex, and BH densities (Extended Data Fig 2). In addition, we have added the R100A and R104A mutants to the biochemical assay (Fig. 3c).

6. The interactions between DNA and RNA backbones are not convincing. Such features are common for low-resolution cryo-EM maps – if one were to go to a sufficient threshold, interactions start to appear. This reviewer would suggest removal of this section.

In the revised cryo-EM map, we confirmed the DNA-RNA backbone contact. We have added the density map of SL2/5 and the heteroduplex (Extended Data Fig 2).

7. The entire region of density should be shown in Ext. Data Fig. 1g, rather than volume around only R323 and Q326.

We have added the entire density of the TAM-recognition panel (Extended Data Fig 2).

8. Please show density for HNHL.

We have added the HNHL density (Extended Data Fig 2).

9. Where is the comparison between Δ SL1 and full-length complex? There is no direct comparison.

We have added the direct comparison between Δ SL1 and full-length complexes (Extended Data Figs. 6e and 6f).

Minor comment:

1. It is not surprising that the RNA makes little contact with targeted DNA, aside from the spacer base-pairing with the target DNA. This is a puzzling parallel to make since it seems rather obvious based on our knowledge of RNA guide - DNA interactions. It's possible that this reviewer is missing something?

We appreciate the Reviewer's point regarding the relevance of RNA-DNA interactions in *IsrB*-mediated DNA targeting. It is true that the RNA guide region and the target DNA make contact through base-pairing; however, in stark contrast to all known CRISPR systems, other non-specific RNA-DNA contacts are demonstrated by our structure as well, showing the importance of the ω RNA in recognizing the RNA/DNA duplex. The base and backbone of C198 in the RNA make multiple contacts between 3 and 5 Å with the phosphate and deoxyribose moieties of the DNA at positions -6 and -7 (relative to the TAM) in the RNA/DNA duplex, demonstrating the ω RNA has evolved at least one RNA-mediated (as opposed to protein-mediated) mechanism to specifically recognize the duplex structure. This observation places *IsrB* and its ω RNA in a category more like that of bacterial self-splicing introns in which the RNA plays a more prominent role than simply providing a guide and delivering the protein adaptor. We have added this discussion to the text to clarify.

David Taylor

Reviewer Reports on the First Revision:

Referees' comments:

Referee #1:

This is the revised version of the manuscript "Structure of RNA-guided nickase IsrB in complex with guide RNA and target DNA".

The authors have provided new data – in particular better cryoEM maps, which was a major criticism of the original submission.

In response to my question around the stability of the variant proteins, the authors provide a new SDS-PAGE gel (Ext. Data 5b) that appears to show partly purified proteins. The figure does not explain how these were purified, and the response to reviewers suggests this figure shows bacterial lysates, which doesn't seem likely. Please provide more detail.

Malcolm White

Referee #2:

The authors have addressed all my comments. I appreciated the resolution improvement of the presented cryo-EM map during revision. This is now a very nice piece of work.

Referee #3:

I appreciate the authors, providing the PDB reports! The authors' structural analysis appear solid. The higher resolution structures, accompanied with a more thorough biochemical analysis, make the story much stronger. One minor comment: the map appears to be over sharpened, perhaps the authors should display the unsharpened map for those who thoroughly analyze EM structures. While the work by Ke and colleagues obviously affect the novelty of this work, I still recommend publication in Nature, or immediate publication in NSMB. This story should still be a great interest to general readers and multiple papers describing the structures, should draw further attention for comparisons.

David Taylor

Author Rebuttals to First Revision:

Referee #1:

This is the revised version of the manuscript “Structure of RNA-guided nickase IsrB in complex with guide RNA and target DNA”.

The authors have provided new data – in particular better cryoEM maps, which was a major criticism of the original submission.

In response to my question around the stability of the variant proteins, the authors provide a new SDS-PAGE gel (Ext. Data 5b) that appears to show partly purified proteins. The figure does not explain how these were purified, and the response to reviewers suggests this figure shows bacterial lysates, which doesn't seem likely. Please provide more detail.

Malcolm White

We apologize for the confusion regarding the SDS-PAGE gel (Ext. Data 5b). You are correct - these are not bacterial lysates, but rather the purified protein following bacterial expression. We have added the following sentences in the method section to describe this: “To examine the protein stability of deletion mutants, IsrB proteins were produced in the bacterial expression system used in the cryo-EM sample preparation. The *E. coli* cells were resuspended in buffer A (50 mM Tris-HCl, pH 8.0, 20 mM imidazole, and 1 M NaCl), lysed by sonication, and then centrifuged. The supernatant was mixed with MagneHis beads (Promega). The protein-bound column was washed with buffer A. The protein was eluted with buffer B (50 mM Tris-HCl, pH 8.0, 0.3 M imidazole, and 1 M NaCl) and analyzed by SDS-PAGE (Extended Data Fig. 5b).”

Referee #3:

I appreciate the authors, providing the PDB reports! The authors' structural analysis appear solid. The higher resolution structures, accompanied with a more thorough biochemical analysis, make the story much stronger. One minor comment: the map appears to be over sharpened, perhaps the authors should display the unsharpened map

for those who thoroughly analyze EM structures. While the work by Ke and colleagues obviously affect the novelty of this work, I still recommend publication in Nature, or immediate publication in NSMB. This story should still be a great interest to general readers and multiple papers describing the structures, should draw further attention for comparisons.

David Taylor

We are grateful for your expertise and comment regarding the cryo-EM map. According to your suggestion, we have added the unsharpened map to Extended Data Fig. 1a.